# Proximity to Photosystem II is necessary for activation of Plastid Terminal Oxidase (PTOX) for photoprotection

Pablo Ignacio Calzadilla [1], Junliang Song[1], Patrick Gallois [2] & Giles Nicholas Johnson [1] ✉

The Plastid Terminal Oxidase (PTOX) is a chloroplast localized plastoquinone oxygen oxidoreductase suggested to have the potential to act as a photoprotective safety valve for photosynthesis. However, PTOX overexpression in plants has been unsuccessful at inducing photoprotection, and the factors that control its activity remain elusive. Here, we show that significant PTOX activity is induced in response to high light in the model species *Eutrema salsugineum* and *Arabidopsis thaliana*. This activation correlates with structural reorganization of the thylakoid membrane. Over-expression of PTOX in mutants of *Arabidopsis thaliana* perturbed in thylakoid stacking also results in such activity, in contrast to wild type plants with normal granal structure. Further, PTOX activation protects against photoinhibition of Photosystem II and reduces reactive oxygen production under stress conditions. We conclude that structural re-arrangements of the thylakoid membranes, bringing Photosystem II and PTOX into proximity, are both required and sufficient for PTOX to act as a Photosystem II sink and play a role in photoprotection.

Photosynthesis is the fundamental process through which plants fix carbon dioxide into sugars, transforming light into chemical energy. This depends on the primary reactions of two photosystems, PSI and PSII, which receive light energy from excited chlorophylls and use that energy to drive electron transfer to electron acceptors, in a process called charge separation. Stabilizing this charge pair requires a flow of electrons along the photosynthetic electron transport chain (PETC). Failures or blockages in this chain can lead to production of reactive oxygen species (ROS)[1].

When electron flow downstream of PSII is limited, charge recombination reactions can occur, producing singlet excited oxygen ($^1O_2^*$). $^1O_2^*$ can cause destruction of the PSII reaction centers, in a process called photoinhibition[2]. In most plants, the main mechanism protecting PSII from photoinhibition is high energy state quenching (qE; a form of non-photochemical quenching)[3]. qE dissipates absorbed light energy as heat, decreasing the probability of charge recombination reactions and, consequently, ROS production. In recent years, an additional photoprotective pathway has been suggested in a small number of plants – electron flow to the Plastid Terminal Oxidase (PTOX)[4–7].

PTOX is a chloroplast localized plastoquinone (PQ) oxygen oxidoreductase, which oxidizes plastoquinol and reduces $O_2$ to produce $H_2O$. PTOX was first identified in mutant plants showing a mottled green phenotype (*immutans* in *Arabidopsis thaliana*; *ghost* in tomato)[8,9]. This phenotype has been linked to its involvement in oxidizing reduced plastoquinone formed during carotenoid biosynthesis[8–12]. It was also speculated that PTOX could act as an electron sink for PSII, mediating pseudocyclic electron flow from water to oxygen, reforming water, and protecting against environmental stress[13–15]. Plants which over-express PTOX were produced in different species[16–18], however, no increase in short-term stress tolerance was observed in the PTOX over-expressor lines. Rather, tobacco over-expressor plants were shown to be *more* vulnerable to environmental stress[18]. Thus, it remains unclear whether PTOX can in fact act in photoprotection.

[1]Department of Earth and Environmental Sciences, Faculty of Science and Engineering, University of Manchester, Manchester, United Kingdom. [2]Faculty of Biology, Medicine and Health, University of Manchester, Manchester, United Kingdom. ✉e-mail: giles.johnson@manchester.ac.uk

Although PTOX overexpression has been unsuccessful at inducing photoprotection, published evidence shows that PTOX is a sink for electrons from PSII in a small number of extremophile species[4–7,19]. *Eutrema salsugineum* (previously *Thellungiella halophila*, hereafter Eutrema)[4,20], a salt tolerant brassica species, shows PTOX-mediated electron transfer from PSII to $O_2$, accounting for up to 30% of total electron flux under salt stress[4]. Similar conclusions have been drawn in the artic-alpine species *Ranunculus glacialis*[5,19], and indirect evidence of high PTOX activity has been seen in other species from extreme environments[6,7]. The mechanism of regulation of PTOX in these diverse species remains unknown.

Different models for PTOX activation have been suggested. First, a pH-dependent activation was proposed, in which an increase of stromal pH under high excitation pressure triggers PTOX association with the membrane, allowing oxidation of the PQ pool[13,21]. Another recent model suggested redox regulation of PTOX through conserved regulatory disulfides[22]. However, neither model explains the substantial fluxes to PTOX seen in extremophile plants under stress conditions.

Previously, we proposed that diffusion of PQ from PSII to PTOX will be the limiting step for PTOX activity[20]. The plant thylakoid membrane is a complex folded structure, divided into two main regions – the grana and the stromal lamellae[23]. PSII is specifically targeted to the grana[24]. This is a densely stacked membrane structure, and it has been shown that the high protein content of the grana limits PQ diffusion[25]. Meanwhile, PTOX was shown to be localized to the stromal lamellae, specifically to the stroma facing side of the membrane[26]. Thus, in most plants, the distance between PSII and PTOX would preclude a significant electron flow, and PTOX would need to be co-located with PSII to function as an electron sink pathway. In Eutrema, we observed that PSII and PTOX

were co-located under salt stress conditions[20], although the mechanisms giving rise to this remain unclear.

Here, we study PTOX activation during high light acclimation in Eutrema. We provide evidence that unstacking of thylakoid membranes is associated with PTOX activation. We show that photoprotective PTOX activity can be induced in Arabidopsis plants when thylakoids are unstacked, and demonstrate that bringing Photosystem II and PTOX into proximity, is sufficient to result in a significant electron flux between these complexes. Our results contribute to strategies to use PTOX as a pathway to increase stress tolerance, redesigning photosynthesis to optimize photoprotection in plants.

## Results

### High light acclimation triggers PTOX activation in *Eutrema salsugineum*

Six-week-old Eutrema plants were subjected to control (Ct, 100 µmol $m^{-2}$ $s^{-1}$) or high light (HL, 800 µmol $m^{-2}$ $s^{-1}$) conditions for 12 days. HL-treated plants significantly increased their maximum photosynthetic capacity ($P_{max}$) after 3 days, as well as their chlorophyll a/b ratio (Fig. 1A; Supplementary Fig. S1), showing that HL triggers photosynthetic acclimation in Eutrema. To measure PTOX involvement in this response, PSII efficiency (ΦPSII) was estimated under normal and low oxygen conditions (21% $O_2$ and 1% $O_2$, respectively; Fig. 1B). A significant reduction in ΦPSII was observed at 1% $O_2$ in HL-treated plants, after 7 days of stress, consistent with PTOX activation (Fig. 1B and Supplementary Fig. S1D). Relative electron transport through PSII (PSII rETR) was estimated under both $O_2$ conditions, and at different light intensities (Fig. 1C). $O_2$ sensitivity was not observed in Ct but was present in HL-treated plants.

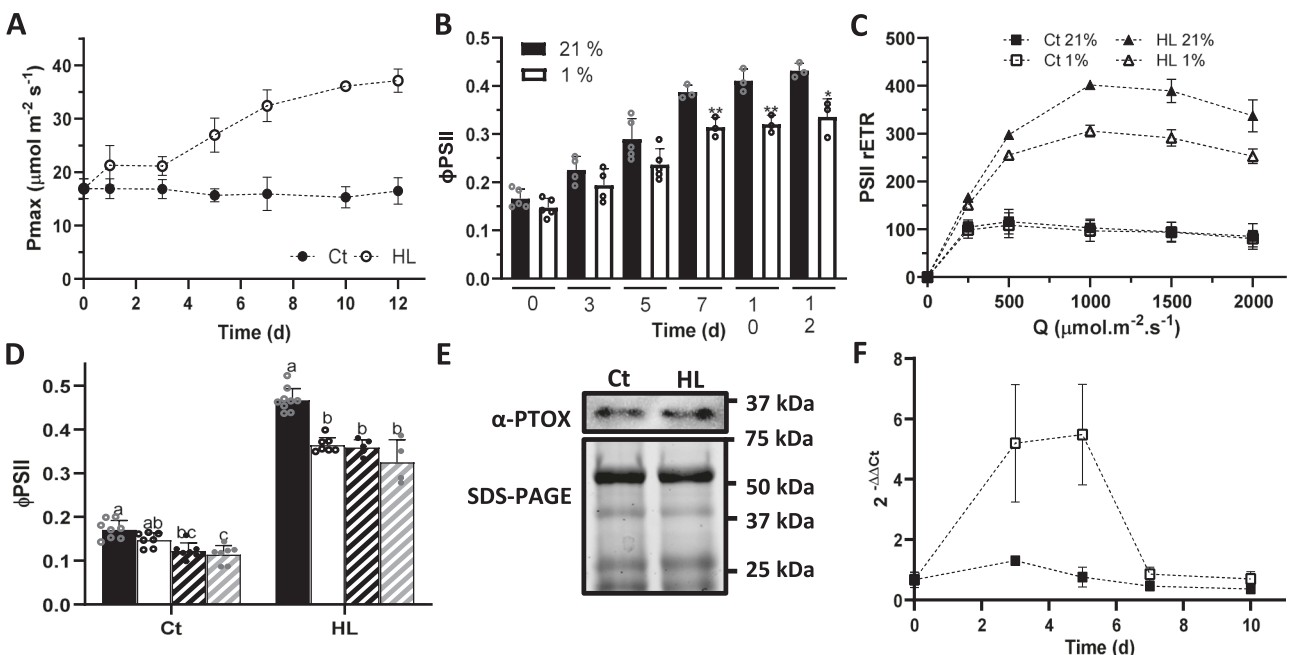

**Fig. 1 | PTOX activity is triggered by high light in *Eutrema salsugineum*.**
**A** Maximum photosynthetic capacity (Pmax) of Eutrema plants subjected to control (Ct) or high light (HL) treatment (*n* = at least 3). **B** Quantum yield of PSII (ΦPSII) measured at 21% or 1% $O_2$ (black and white bars, respectively) during HL acclimation (*n* = at least 3). The asterisk indicates differences between measurements at 21 and 1 % $O_2$ (Student's *t* test, *p* < 0.05; **p* < 0.01, two-sided). **C** Relative electron transport rate of PSII (PSII rETR) at 21% or 1% $O_2$ (filled and empty symbols, respectively), in 10 d Ct or HL-treated plants (*n* = 3). **D** Measurements of ΦPSII in vacuum-infiltrated Eutrema leaves of 10 d Ct or HL-treated plants. Black bars, $H_2O$ infiltration and 21% $O_2$; white bars, $H_2O$ infiltration and 1% $O_2$; black stripped bars, 1 mM nPG and 21% $O_2$; gray stripped bars, 1 mM nPG and 1% $O_2$ (*n* = at least 4). Different letters above

the error bars indicate statistically different values (ANOVA, Tukey's test, *p* < 0.05). **E** PTOX protein content in 10 d Ct and HL-treated plants (protein loaded = 100 µg). Immunoblots were repeated twice, independently, obtaining the same results. **F** Transcript Expression analysis of PTOX in Ct and HL-treated plants (filled and empty symbols, respectively). Gene quantification was determined based on the expression of the PTOX gene relative to the housekeeping gene GADPH (*n* = at least 3). Data are the mean ± SD of independent biological replicates. All raw data, with number of replicates per data point, are provided in the source data file. Measurements conditions of **A**, **B** and **D** were 2000 ppm $CO_2$ and 1000 µmol.$m^{-2}$.$s^{-1}$ light. **C** was measured at 2000 ppm $CO_2$ and different light intensities.

Oxygen sensitivity of electron transport could be due to PTOX activity, or other processes such as photorespiration or the Mehler reaction (photoreduction of $O_2$ by PSI). Measurements were performed at 2000 ppm $CO_2$, so we can exclude substantial photorespiration[4]. However, as we cannot exclude a contribution of Mehler reaction, we tested the effect of the PTOX inhibitor n-propyl gallate (nPG) on $\Phi$PSII at Day 10 of treatments. Leaf infiltration with nPG decreased $\Phi$PSII in HL-treated leaves to a similar extent to low oxygen, confirming activation of PTOX (Fig. 1D). PTOX protein content was estimated at Day 10 to address whether the activity measured was correlated with its protein content. No differences were observed in PTOX protein content between treatments (Fig. 1E). Therefore, the activity of PTOX in the HL-treated plants cannot simply be explained by an increase in its relative protein concentration. PTOX transcript levels were measured to determine whether high light acclimation triggered any regulation in gene expression (Fig. 1F). A significant increase in PTOX transcript was observed in the HL-treated plants at Days 3 and 5, were PTOX activity was not observed, suggesting that PTOX might have a different role at shorter times of HL acclimation.

## Chloroplast structural re-organization is associated with PTOX activation

Based on our previous results in salt-stressed Eutrema[20], PTOX activation depends on processes bringing PSII and PTOX close together. Since these mechanisms of activation, and their molecular pathways, remain unknown, we aimed to identify candidate genes and/or processes for PTOX activation using transcriptomics. RNA samples were taken from Eutrema leaves at Days 0, 3, 5 and 10 of HL treatment (Fig. 1B–D). In total, 6233 genes were significantly differentially expressed between Days 0 and 3, 1073 between Days 3 and 5, and 1770 between Days 5 and 10 ($p_{adj} < 0.1$; Fig. 2A–C). Of these, 4010 genes ($p_{adj} < 0.01$) were classified into five different clusters of gene expression profiles (Fig. 2D; Supplementary Fig. S2), for which biological significance was assigned through a GO enrichment analysis ($p_{adj} < 0.05$; Supplementary Figs. S3–5).

Given the chloroplast localization of PTOX, we focused our attention on GO terms in the 'Cellular Compartment' GO classification group associated with the chloroplast (Fig. 2E and Supplementary Fig. S5). In total, 152 genes were selected within this group, all belonging to Clusters 1 and 2, for which we assessed their molecular function (Fig. 2F–M; Supplementary Table S1). Of these, 62 were related to stress response mechanisms and to different photosynthetic processes – cyclic electron flow (CEF), photorespiration and the Calvin–Benson–Bassham (CBB) cycle (Fig. 2F–I; Supplementary Table S1). The remaining clustered genes (90) were related to protein turnover, protein translocation, PETC biogenesis and chloroplast structural organization (Fig. 2J–M).

Considering the PSII-PTOX proximity model, PTOX could gain access to PSII via the thylakoid lumen, which would imply the participation of a translocation system, or through changes in thylakoid structure[20]. Although we found genes belonging to the TAT and SEC-translocation systems up-regulated by HL, suggesting an increase in protein lumen-translocation triggered by stress (Fig. 2K and Supplementary Table S1), PTOX is not predicted to be translocated to the thylakoid lumen based on targeting protein signals (Supplementary Fig. S6). We did however identify an increase in the expression of genes triggering changes in thylakoid membrane architecture, including CURT1A (Thhalv10029019m), FZL (Thhalv10006729m), RIQ1 and RIQ2 (Thhalv10014904m and Thhalv10019168m, respectively)[27–30] (Fig. 2M and Supplementary Table S1). This is consistent with chloroplast structural re-organization due to HL[31,32]. Thus, we decided to study chloroplast ultrastructure changes in Eutrema to determine the extent of structural re-organization in this extremophile species.

HL-acclimated leaf samples were taken, and their chloroplast ultrastructure analyzed using transmission electron microscopy (TEM)

(Fig. 3A). We defined a grana stack as three or more thylakoid membranes stacked together, as previously defined by Mazur et al.[33]. No differences were observed in the number of grana stacks per chloroplast between Days 0 and 10 (Fig. 3B), however the number of thylakoid layers per grana and the total grana height significantly decreased in HL (Fig. 3C, D). An increase in the diameter of the grana stacks was observed between Day 0 and 10 (Fig. 3E). These chloroplast structural re-arrangements were not a consequence of changes in the developmental stage of the leaves (Supplementary Fig. S7), and altogether, showed that HL triggered reduced stacking of the thylakoid membranes in Eutrema. This phenomenon occurred over the course of the HL treatment and correlated with the kinetics of PTOX activation (Figs. 1 and 3A). Consequently, we hypothesized that thylakoid unstacking participates in PTOX activation at HL.

HL is known to cause a decrease in grana stacking in many plant species[31,32], suggesting that this may give rise to PTOX activity. To confirm whether responses of Arabidopsis differed to those of Eutrema, we exposed Arabidopsis to HL or Ct treatments for 12 days and measured PTOX activity and chloroplast ultrastructure changes (Fig. 3F–M). We observed a small but significant PTOX activity under HL (Fig. 3H, I), together with a decrease in the number of thylakoids per grana and the total grana height (Fig. 3K, L). Thus, thylakoid unstacking is also associated with PTOX activity under HL in Arabidopsis. However, the structural changes in Arabidopsis contrasted with those in Eutrema. Arabidopsis had more grana stacks under Ct conditions than Eutrema, and that number increased further under HL (Fig. 3J c.f. B). At the same time, granal diameter decreased at HL in Arabidopsis, whilst increasing in Eutrema (Fig. 3M c.f. E). Nevertheless, the total length of thylakoid membranes appressed within grana stacks was significantly reduced by HL, both in Eutrema and Arabidopsis plants (Supplementary Fig. S8).

HL has also been reported to induce grana margin expansions and swelling of the peripheral grana, which has been suggested to facilitate the PSII repair cycle[34,35]. However, none of these phenomena were observed under our experimental conditions, in either of the species studied (Supplementary Fig. S9).

## Overexpression of EsPTOX in Arabidopsis thylakoid stacking mutants confers PTOX activity

To test the impact of thylakoid stacking on PTOX regulation, we over-expressed Eutrema salsugineum PTOX (EsPTOX) in Arabidopsis mutants with defective thylakoid stacking[36]. chl1-3 is a chlorophyll b-less mutant, giving it a pale-green appearance, with a reduced number of grana stacks per chloroplast compared to wild type (wt); while chl1-3xlhcb5 (additionally mutant in the LHCII Chl a/b protein 5) has a more pronounced phenotype than chl1-3 (Fig. 4)[36]. These mutants have previously been shown to have reduced photosynthetic performance and increased sensitivity to photodamage[36].

PTOX activity was assayed by rETR sensitivity to oxygen and nPG infiltration, in the Arabidopsis wt, chl1-3 and chl1-3xlhcb5 lines (Fig. 4A–D, Supplementary Fig. S10). No intrinsic PTOX activity was observed in these genetic backgrounds when EsPTOX was not over-expressed. However, over-expression of EsPTOX induced rETR oxygen sensitivity in these stacking mutants but not in the wt background (Fig. 4). Similarly, nPG infiltration reduced $\Phi$PSII only in the chl1-3-EsPTOX and chl1-3xlhcb5-EsPTOX lines, confirming that this effect was due to EsPTOX expression, and that the ability of EsPTOX to act as a PSII electron sink is associated with the impairment in thylakoid stacking. Consistent with this, no oxygen sensitivity of rETR was observed in the lhcb5 or lhcb5-EsPTOX genetic backgrounds, which are unaffected in thylakoid stacking (Supplementary Fig. S11).

We further explored the effect of EsPTOX expression on the photosynthetic capacity of the wt, chl1-3 and chl1-3xlhcb5 genetic backgrounds. Rates of $CO_2$ assimilation (A) and $\Phi$PSII were measured under growth conditions (400 ppm $CO_2$ and 100 $\mu$mol m$^{-2}$ s$^{-1}$ light),

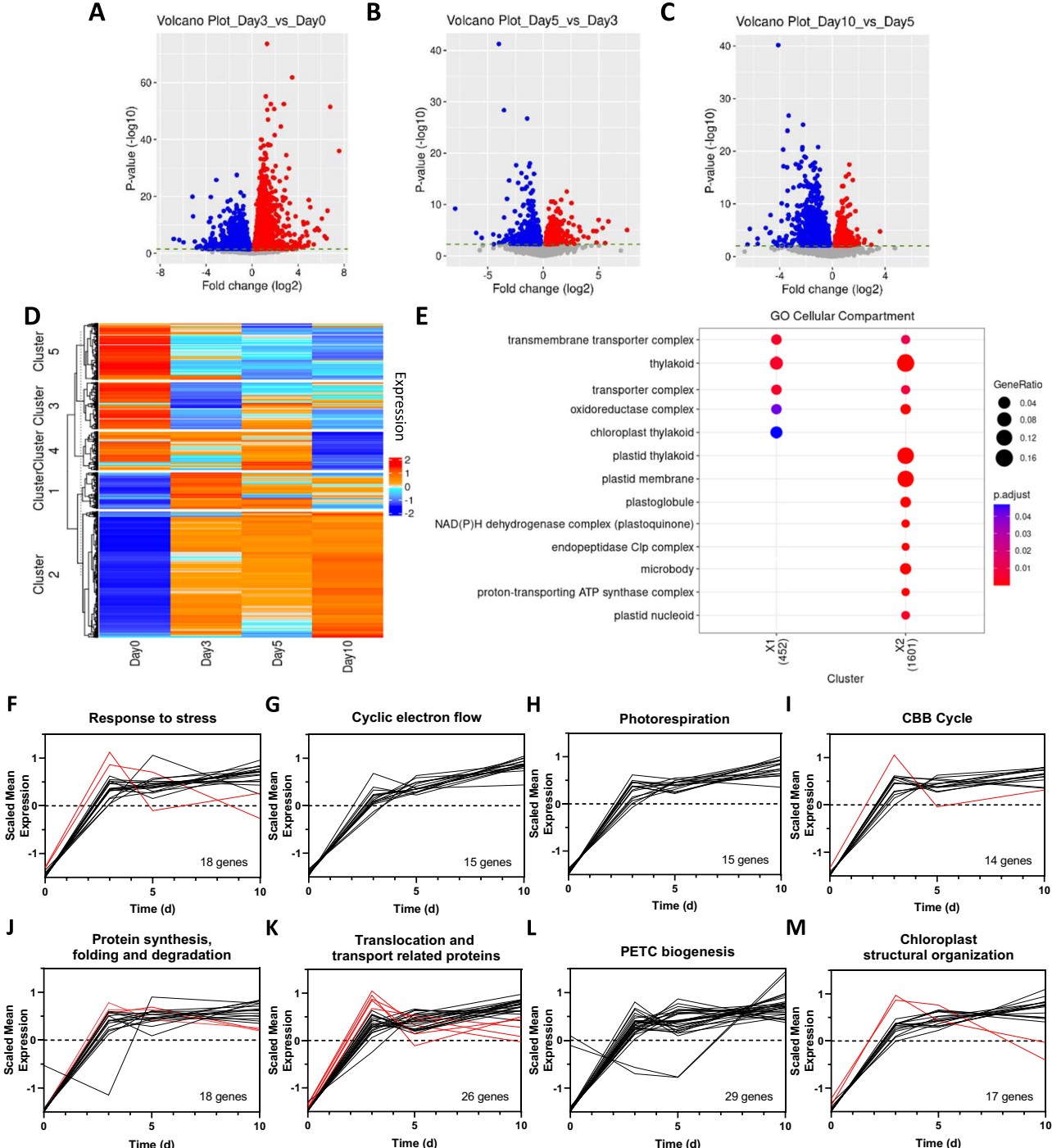

**Fig. 2 | Gene expression changes induced by high light in *Eutrema salsugineum*.** **A−C** Volcano plots showing statistically significant changes in gene expression and their fold-change during the time-course RNAseq experiment. Blue and red dots correspond to significant differentially expressed genes ($p_{adj}$ <0.1) with a fold change ($log_2$) < 0 or >0, respectively. DESeq2 was used for the normalization and to generate the *p*-values using their default Wald test. **D** Clustering of the differentially expressed genes along the high light (HL) treatment in Eutrema. **E** GO-enriched terms related to chloroplast localization in the 'Cellular Compartment' classification category of *Gene Ontology*. GO enrichment was performed with clusterProfiler_4.4.4 and the TAIR database org.At.tair.db_3.15.1. **F−M** Functional classification of the genes enriched in GO terms related to *chloroplast* in the 'Cellular Compartment' category of *Gene Ontology*. Red and black lines correspond to genes grouped in Cluster 1 and 2, respectively.

and at saturating $CO_2$ (2000 ppm $CO_2$ across different irradiances) (Supplementary Fig. S12 and Fig. 5, respectively). Expressing EsPTOX in the wt background did not significantly affect A or electron transport rates in either condition. In the *chl1-3* and *chl1-3xlhcb5* backgrounds, EsPTOX over-expression resulted in plants that had a lower assimilation at high light and $CO_2$ (i.e. reduced $P_{max}$; Fig. 5D−F).

To address whether this reduction in $P_{max}$ was an effect of PTOX activity competing with $CO_2$ assimilation, we estimated $P_{max}$ under PTOX-inactive conditions (1% $O_2$) in the different genetic backgrounds (Fig. 5G−I). Pmax was insensitive to $O_2$ in all lines. Thus, the reduction of $P_{max}$ in the EsPTOX stacking mutants was not due to PTOX directly competing with carbon fixation. Rather, we conclude, that PTOX

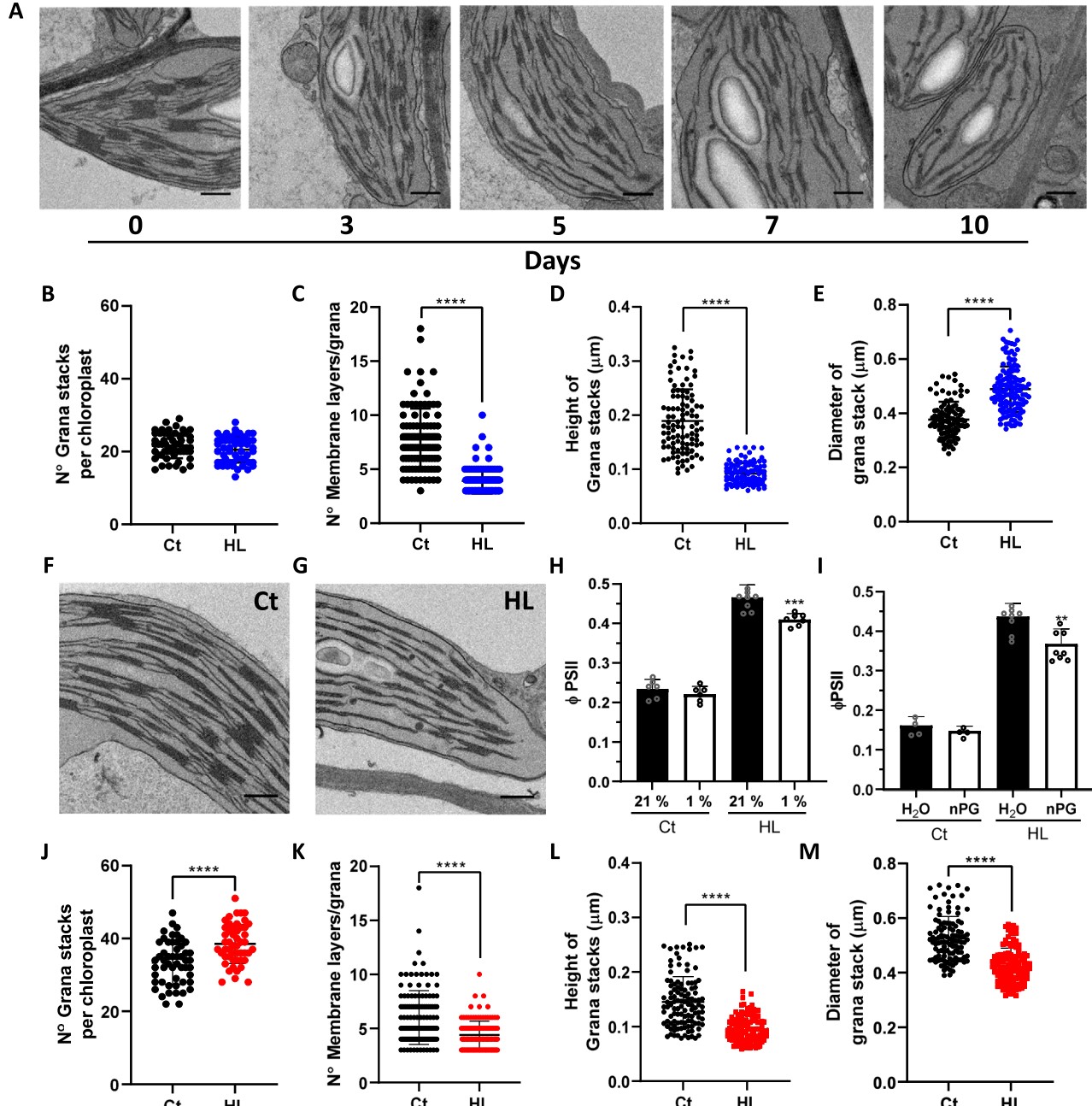

**Fig. 3 | Chloroplast structural re-organization is associated with PTOX activation. A** Micrographs of Eutrema leaves exposed to different days of high light (HL) treatment. **B–E** Chloroplast ultrastructure parameters calculated at 0 (Ct) and 10 d of HL in Eutrema. **F, G** Micrographs of Arabidopsis leaves exposed to Ct or HL, respectively, for 12 d. **H** ΦPSII measured at 21% or 1% $O_2$, in Arabidopsis plants exposed to 12 d Ct or HL conditions. **I** ΦPSII was measured in vacuum-infiltrated leaves of 12 d Ct or HL-treated Arabidopsis plants. Infiltration was performed with water or 1 mM nPG. H and I. were measured at 1000 μmol m$^{-2}$ s$^{-1}$ light and 2000 ppm $CO_2$ **J–M** Chloroplast ultrastructure parameters calculated for Ct and HL Arabidopsis. Data are the mean ± SD of at least $n = 50$ chloroplasts (**B, J**) or $n =$ at least 100 grana stacks (**C–E, K–M**), and at least 4 biological replicates for **H** and **I**. A minimum of 25 non-overlapping fields of view of the ultra-thin sections were examined, and leaf pieces of two plants were fixed for the analysis. The asterisk indicates differences between Ct and HL treatments for **B–E** and **J–M** (Welch's test, ****$p < 0.0001$, two-sided) and differences between 21% and 1% $O_2$, or $H_2O$ and nPG for **H** and **I**, respectively (Student's $t$ test, **$p < 0.01$, ***$p < 0.001$, two-sided). All raw data, with number of replicates per data point, are provided in the source data file. Scale bars = 0.5 μm.

activity alters photosynthetic acclimation through changes in the redox state of the PETC[37,38].

## PTOX activity reduces photoinhibition and ROS generation in Arabidopsis thylakoid stacking mutants

Previous studies have suggested that over-expressed PTOX is not photoprotective in Arabidopsis, and that it actually increases photoinhibition in tobacco plants[17,18]. These conclusions were obtained by

following photoinhibition, estimated as changes in the maximum quantum yield of PSII (Fv/Fm), of leaves exposed to high light and cold stress, or lincomycin (a protein synthesis inhibitor). To address whether PTOX was photoprotective in the stacking mutant backgrounds, we repeated the photoinhibition experiment previously described[17]. Leaves of wt, *chl1-3* and *chl1-3xlhcb5* were exposed to high light and cold (1000 μmol m$^{-2}$ s$^{-1}$ and 4 °C) for a total of 8 h, and Fv/Fm was measured every 2 h (Fig. 6). Measurements were compared to control

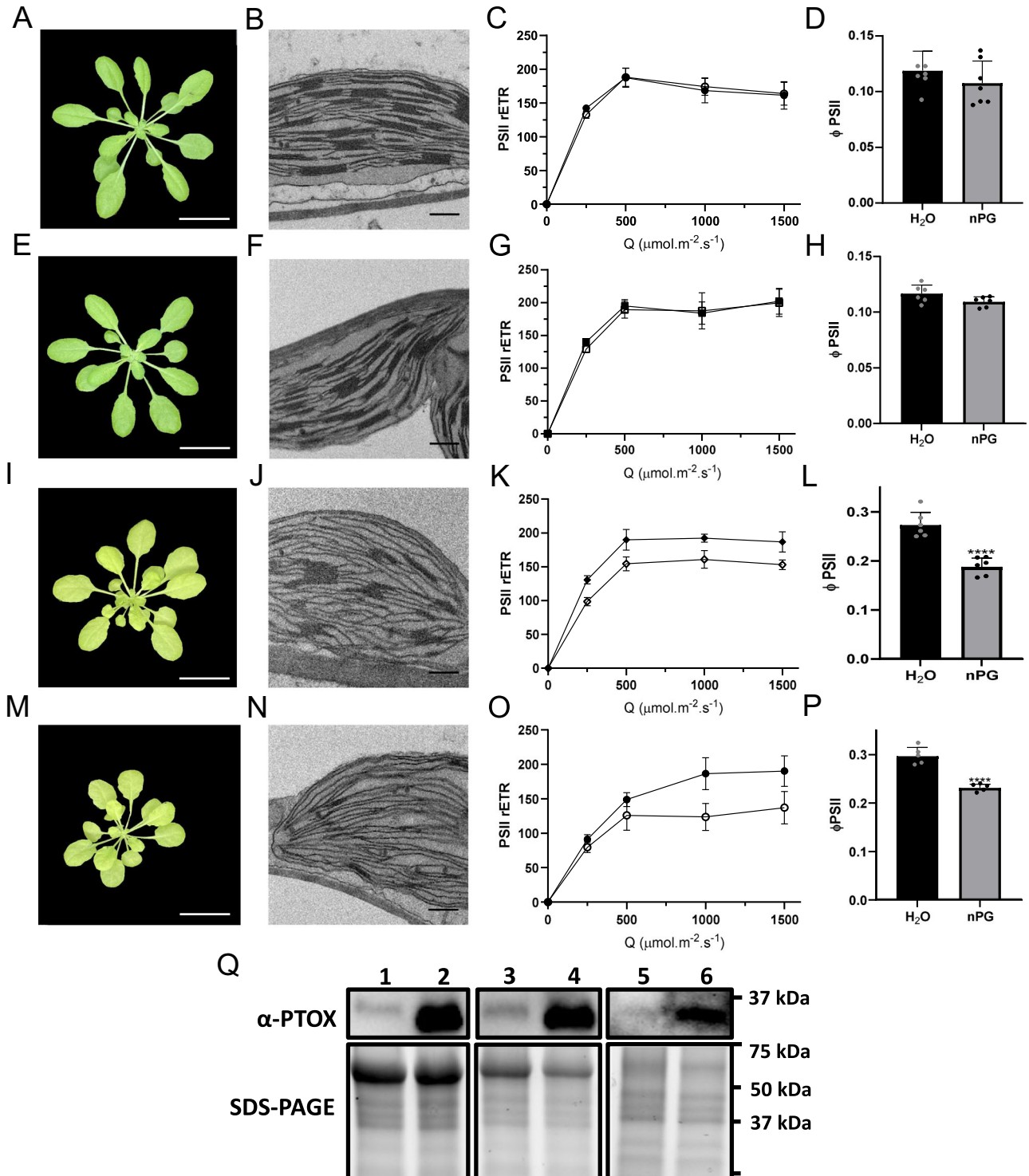

**Fig. 4 | PTOX activity in Arabidopsis thylakoid stacking mutants expressing EsPTOX.** Measurements were performed in Arabidopsis wt (**A**–**D**), wt-EsPTOX (**E**–**H**), *chl1-3*-EsPTOX (**I**–**L**) and *chl1-3xlhcb5*-EsPTOX (**M**–**P**) genetic backgrounds. Representative images of 5-week-old wt (**A**) and EsPTOX (**E**), 8-week-old *chl1-3*-EsPTOX (**I**) and 10-week-old *chl1-3xlhcb5*-EsPTOX (**M**). Chloroplast ultrastructure was analyzed by transmission electron microscopy (TEM) (**B**, **F**, **J**, **N**; scale bars = 0.5 μm). Relative electron transport rate of PSII (PSII rETR) were measured at different light intensities, under 2000 ppm $CO_2$ and at 21% or 1% $O_2$ (filled and empty symbols, respectively)(**C**, **G**, **K**, **O**) (*n* = at least 3). ΦPSII was measured in vacuum-infiltrated leaves at 1000 μmol m$^{-2}$ s$^{-1}$ light and 2000 ppm $CO_2$ (**D**, **H**, **L**, **P**) (*n* = at least 5). Black bars, $H_2O$; gray bars, 1 mM nPG. The asterisk indicates differences between infiltration with $H_2O$ and nPG (Student's *t* test, *p* < 0.0001, two-sided). **Q** PTOX protein content in the different Arabidopsis genetic backgrounds. 1, wt; 2, wt-EsPTOX; 3, *chl1-3*; 4, *chl1-3*-EsPTOX; 5, *chl1-3xlhcb5*; 6, *chl1-3xlhcb5*-EsPTOX. (loaded protein = 60 μg). Immunoblots were repeated twice, independently, obtaining the same results. White scale bars in **A**, **E**, **I** and **M** = 2 cm. Data are the mean ± SD of independent biological replicates. All raw data, with number of replicates per data point, are provided in the source data file.

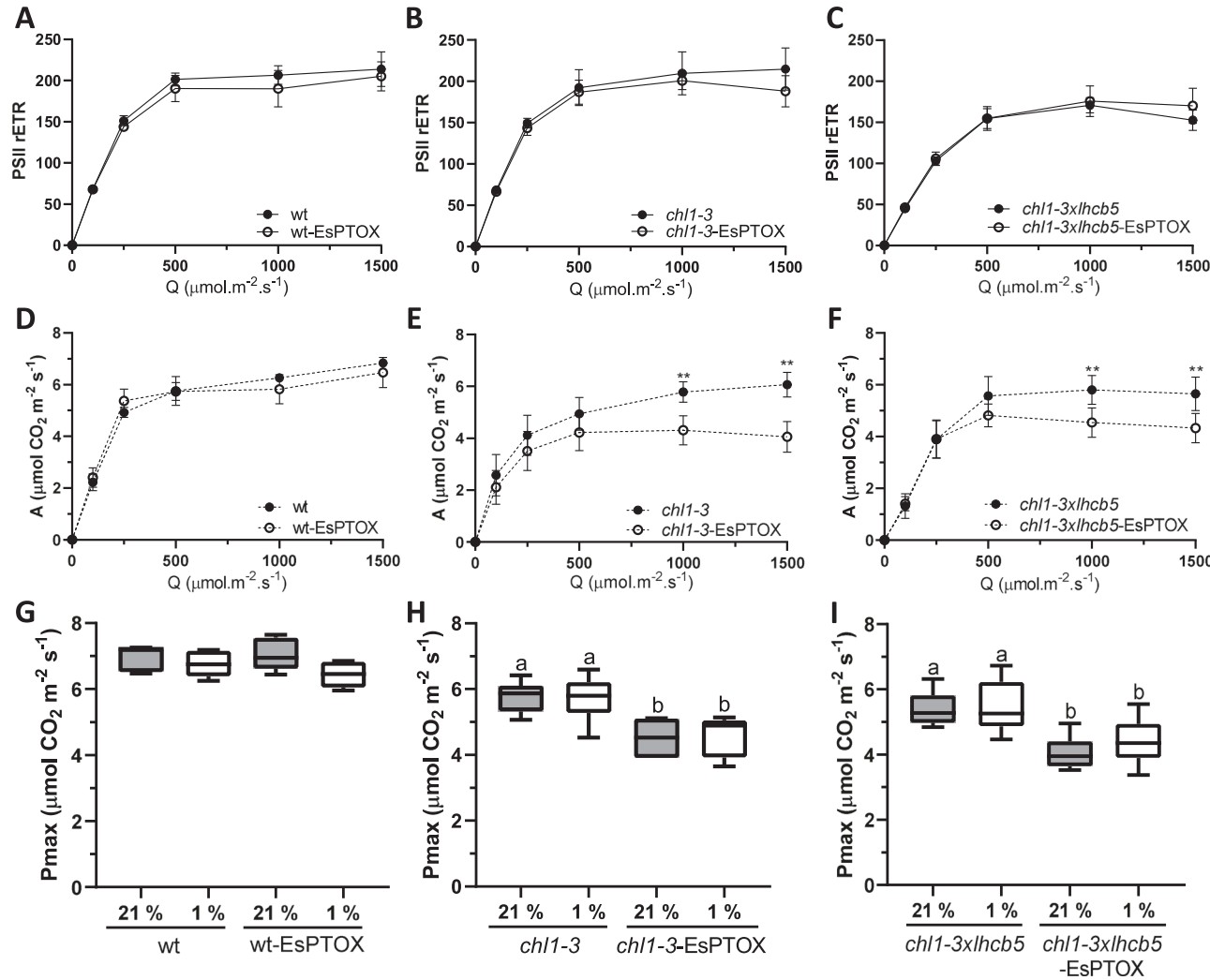

**Fig. 5 | The effect of EsPTOX expression on photosynthetic assimilation rates at saturating $CO_2$ conditions in the Arabidopsis thylakoid stacking mutants.** Relative electron transport rate of PSII (PSII rETR) and assimilation (A, µmol $CO_2$.m$^{-2}$.s$^{-1}$) of wt and wt-EsPTOX (**A, D**), *chl1-3* and *chl1-3*-EsPTOX (**B, E**), and *chl1-3xlhcb5* and *chl1-3xlhcb5*-EsPTOX (**C, F**). Measurements were performed at 2000 ppm $CO_2$ (n = at least 4). The asterisk indicates differences between lines expressing EsPTOX or not, at a certain light intensity (Student's *t* test, *p* < 0.01, two-sided). The maximum capacity for photosynthesis (Pmax) measured under 21% and 1% $O_2$, and irradiance 1000 µmol.m$^{-2}$.s$^{-1}$, in the wt and wt-EsPTOX genotypes (**G**), *chl1-3* and *chl1-3*-EsPTOX (**H**) and *chl1-3xlhcb5* and *chl1-3xlhcb5*-EsPTOX (**I**) (n = at least 4). Data are the mean ± SD of independent biological replicates. Box plots represents the median, lower and upper quartile, while the whiskers the smallest and largest value. Different letters above the error bars indicate statistically different values (ANOVA, Tukey's test, *p* < 0.05). All raw data, with number of replicates per data point, are provided in the source data file.

leaves maintained at 10 µmol m$^{-1}$ s$^{-1}$ and 20 °C. As expected, no differences in photoinhibition rates were observed between the Arabidopsis wt and wt-EsPTOX (Fig. 6A). However, EsPTOX expression reduced photoinhibition under stress in the *chl1-3* and *chl1-3xlhcb5* backgrounds (Fig. 6B, C).

To study the role of EsPTOX in wider photoprotection, ROS levels were estimated using a ROS sensitive dye, dichlorofluorescein diacetate (DCF-DA), after 2 h of either stress or control treatments (conditions as above). Stress induced a similar ROS burst in the wt and wt-EsPTOX, but over-expressing EsPTOX reduced ROS levels in the *chl1-3* and *chl1-3xlhcb5* backgrounds (Fig. 6D–G). This effect was confirmed using confocal microscopy, showing the co-localization of the yellow and magenta fluorescence (DCF-DA and chlorophyll autofluorescence, respectively) at a subcellular level (Supplementary Figs. S13 and S14). No yellow fluorescence was observed in the leaves when the infiltration was performed without the dye (Supplementary Figs. S15 and S16). Our results show that the oxidative burst was generated in the chloroplast,

and that EsPTOX expression was photoprotective and reduced photoinhibition when the thylakoids were unstacked.

## Discussion
Since its first identification, PTOX has been seen as a protein with potential for improving plant stress tolerance[4,5,14,17]. So far though, attempts to induce photoprotective activity by over-expressing PTOX have failed and in one case over-expressing PTOX even increased stress sensitivity[17,18]. Here, we have conclusively shown that PTOX can protect plants against photoinhibition and oxidative stress. We have demonstrated that the barrier to PTOX acting as a significant PSII acceptor is the proximity of these complexes. We have also shown that PTOX activity can be acquired by re-organization of the thylakoid membrane, without the need for a co-factor or a protein partner.

PTOX's role in photoprotection depends on the expression of the protein and its ability to access the pool of PQ molecules in the vicinity of PSII. Over-expressing EsPTOX in a wt Arabidopsis background does

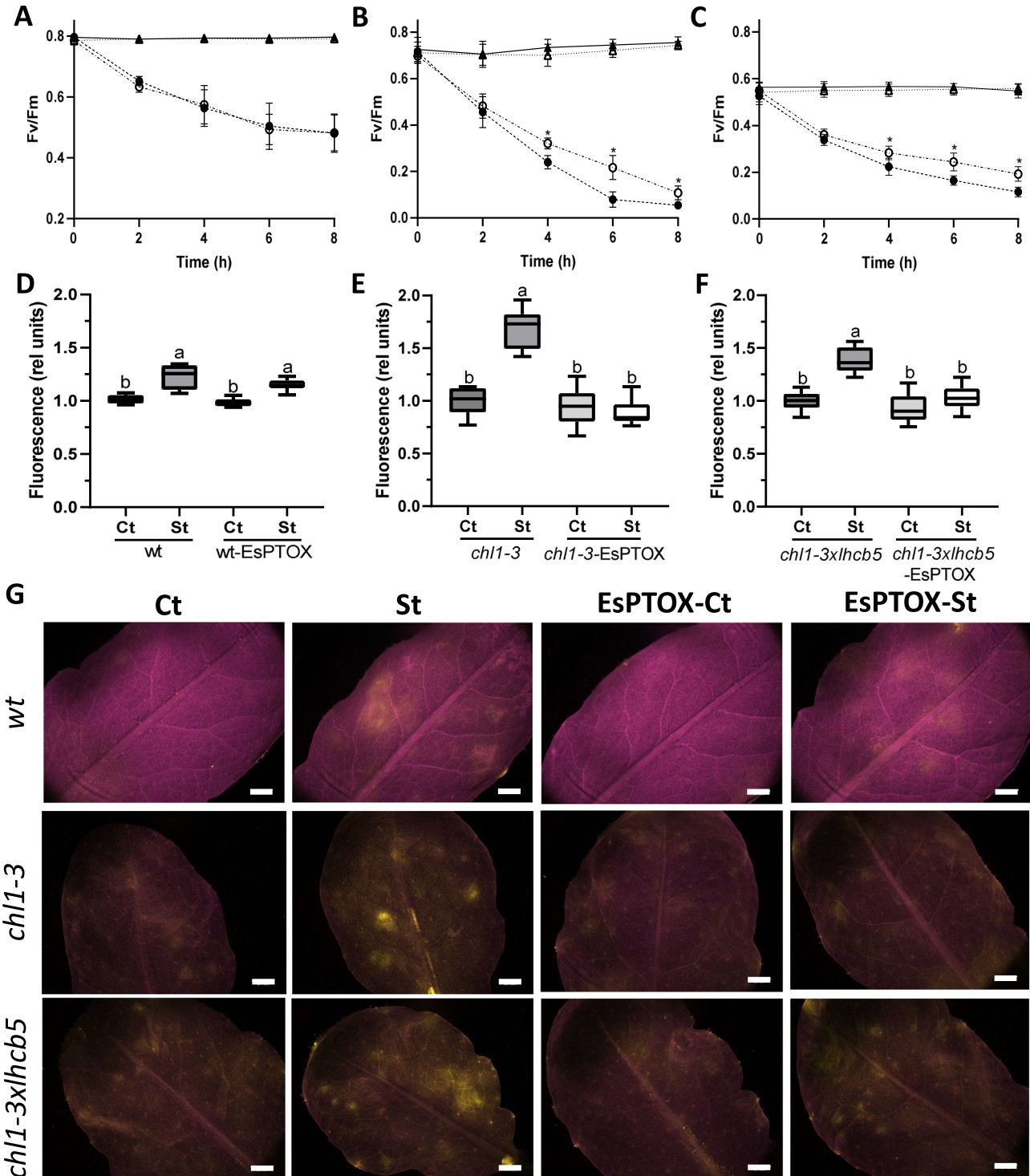

**Fig. 6 | EsPTOX expression reduces photoinhibition and ROS generation in the Arabidopsis thylakoid stacking mutants.** Fv/Fm was recorded in leaves of the wt (**A**), *chl1-3* (**B**) and *chl1-3xlhcb5* (**C**) genetic backgrounds, during 8 h of the photoinhibition (1000 μmol m⁻² s⁻¹ light and 4 °C, circles, St) or control treatment (10 μmol m⁻² s⁻¹ light and 20 °C, triangles, Ct). Filled symbols, no EsPTOX; empty symbols, EsPTOX expressor lines. Data are the mean ± SD of at least 12 biological replicates. The asterisk indicates significant differences between the line expressing EsPTOX or not, under the same treatment (Student's *t* test, *p* < 0.05, two-sided). **D**–**F** ROS fluorescence quantification of leaves subjected to the photoinhibition treatment for 2 h, infiltrated with DCF-DA and then exposed 10 min to 1000 μmol m⁻²s⁻¹ light. **D** wt; **E** *chl1-3*; **F** *chl1-3xlhcb5* genetic backgrounds. Data are the mean ± SD of at least 8 biological replicates. Box plots represents the median, lower and upper quartile, while the whiskers the smallest and largest value. Different letters above the error bars indicate statistically different values (ANOVA, Tukey's test, *p* < 0.05). Values were relativized to the non-expressor genotype under control conditions, in each case. **G** Representative images of the ROS fluorescence (yellow) in the leaves of the different genetic backgrounds. Magenta corresponds to chlorophyll autofluorescence. Scale bar = 10 mm. All raw data, with number of replicates per data point, are provided in the source data file.

not result in PTOX activity per se as its access to PSII is limited (Fig. 4). By contrast, when accessibility is not restrained, as in the stacking mutants, PTOX over-expression leads to activity as a significant PSII electron sink.

In Eutrema, HL induces a substantial reduction in grana stacks and re-organization of the thylakoid membrane (Fig. 3). Similar chloroplast structural changes have been observed in salt-treated Eutrema plants[39], where PTOX activity has also been reported[20]. This thylakoid membrane re-organization is the consequence of a complex regulatory gene network, triggered as an acclimation response. This might explain why PTOX activity is usually observed under long-term[4–7,19] but not short-term stress conditions[17,18,40]. HL-induced thylakoid unstacking is seen in other species, including Arabidopsis[31,32]. Here, we observe that HL also gives rise to a small but significant PTOX activity in wt Arabidopsis (Fig. 3). There are however differences in the structural changes seen at HL between these species, with Eutrema showing fewer grana stacks per chloroplast than Arabidopsis, with those stacks increasing in diameter under HL (Fig. 3). This variation may underlie the differences in PTOX activity seen. A better understanding of the drivers of these structural changes may help identify ways to increase PTOX activity in non-stress tolerant plants.

PTOX is ubiquitous in plants, and mutants such as *immutans* and *ghost* show extreme developmental phenotypes[8,9]. These phenotypes have been explained by PTOX acting as an electron sink during carotenoid synthesis[12]. That PTOX can act as a PSII electron sink suggests a broader role in the leaf. During chloroplast development, chlororespiration, involving electron flow via NDH and PTOX may provide ATP[41], while, in immature chloroplasts, with less thylakoid stacking, PSII-PTOX electron transport may be important in protecting PSII during its assembly. In stress-tolerant plants, this pathway is co-opted as a stress-induced photoprotective process. We can speculate that PTOX may also act as an electron sink during the PSII repair cycle[42], when PSII is reassembled outside the grana. Finally, PTOX may be important during senescence, when again thylakoid stacking breaks down[43,44].

Although PTOX can act as a safety valve for photoprotection, depending on its proximity to PSII, the presence of another regulatory pathway in PTOX activation process cannot be discarded. Since PTOX can affect the redox state of the PQ pool, having an impact on gene expression and acclimation responses[37,38], it is reasonable to hypothesize a secondary mechanism controlling PTOX translocation. This possibility is supported by the dynamics of re-location of a PTOX-GFP fused protein in chloroplast of Arabidopsis, and under different light conditions[45]. PTOX could change its distribution in the thylakoid membranes through specific tether proteins, such as it was recently described for the ferredoxin:NADP(H) oxidoreductase (FNR)[46]. The existence of a putative PTOX carrier protein should be addressed in future studies.

In conclusion, here we have shown that structural re-arrangements of the thylakoid membranes can act as a regulatory step in PTOX activation. We have demonstrated that proximity to PSII is sufficient to give rise to substantial rates of PTOX activity and that that activity is photoprotective. Our data confirm the role of PTOX as a safety-valve for photoprotection and allow us to hypothesize its involvement in other physiological processes where thylakoid membrane organization is affected. This work provides information which may inform approaches to increase stress tolerance by redesigning photosynthesis in plants.

## Methods
### Plant material and high light acclimation treatment
Seeds of *Arabidopsis thaliana* (Col-0, refer as wt) and *Eutrema salsugineum* (Shandong wt), together with the Arabidopsis mutant plants *chl1-3*, *lhcb5* and *chl1-3xlhcb5*, were stratified at 4 °C for 2 days, and then germinated in a controlled-environment cabinet (E. J. Stiell) in an 8 h photoperiod (100 µmol m$^{-2}$ s$^{-1}$ provided by led tubes, warm white, color temperature 3200 K) at 22 °C/18 °C (day/night). 6-week-old

Eutrema or 5-week-old Arabidopsis plants were used for the high light acclimation experiments. High light treatment was performed by placing the plants at 800 µmol m$^{-2}$ s$^{-1}$, in an 8 h photoperiod at 22 °C/18 °C (day/night), for 12 days. The experiments performed to study the Arabidopsis thylakoid stacking mutants, were carried out with 5-week-old wt and *lhcb5* plants, 8-week-old *chl1-3* and 10-week-old *chl1-3xlhcb5*. *chl1-3* seeds were kindly given by Dr. Krishna Niyogi, while *lhcb5* seeds were obtained from NASC seeds collection (ID: N656198). The *chl1-3xlhcb5* line was obtained by crossing homozygous *lhcb5* and *chl1-3* plants. F3 generation seeds of *chl1-3xlhcb5* were used in all experiments.

### Plant transformation and genotyping
Transgenic plants expressing the PTOX gene from *E. salsugineum* (EsPTOX) were generated as described by Stepien and Johnson[20]. Briefly, pENTR1A containing EsPTOX was recombined into pH2GW7 destination vector, via LR clonase reaction, using the Gateway technology (Invitrogen, MA, USA). Expression of EsPTOX was driven under control of the cauliflower mosaic virus (CaMV) 35S constitutive promoter. The resultant pH2GW7-EsPTOX expression vector (Supplementary Fig. S17A) was introduced by the liquid nitrogen freeze-thaw method into *Agrobacterium tumefaciens* strain GV3101, and grown in medium supplemented with antibiotics to select for positive colonies. Then, PCR-verified Agrobacterium colonies were used for transformation of Arabidopsis wt, *chl1-3*, *lhcb5* and *chl1-3xlhcb5*, by floral dipping[47]. Plants were left to flower and seeds were harvested, surface-sterilized and sowed in half-strength Murashige and Skoog (MS) selective medium (50 mg.L$^{-1}$ hygromycin) for the selection of transgenic lines. Long roots seedlings were selected and transferred to 7.5-cm pots filled with Levington F2 compost (Levington Advance, UK). Presence of the EsPTOX gene was confirmed by PCR, and its expression estimated by quantitative PCR (qPCR; Supplementary Fig. S17B–E). Primers used are listed on Supplementary Table S2. Homozygosity for the *lhcb5* backgrounds was checked by PCR (Supplementary Fig. S18), while homozygosity for *chl1-3* was checked by chlorophyll extraction (Supplementary Fig. S19). Mutant plants lacking *chl1-3* do not synthesize chlorophyll *b*[36]. Two independent EsPTOX expressor lines were selected for each genetic background and phenotyped using rETR oxygen sensitivity (see *Measurements of PTOX activity*) (Supplementary Fig. S18). Since the EsPTOX lines in each genetic background showed a similar behavior, only one line was selected in each case for further experiments (particularly, wt-EsPTOX-5; *lhcb5*-EsPTOX-4; *chl1-3*-EsPTOX-6 and *chl1-3xlhcb5*-EsPTOX-F).

### RNA extraction and expression analysis by qPCR
Total RNA was extracted using a Plant Spectrum Total RNA Kit (Sigma, MO, USA) according to the manufacturer's instructions. RNA was checked for quality and quantified using agarose gel electrophoresis and spectrophotometric analysis. Samples were treated with DNase I (Amp Grade, Invitrogen MA, USA) according to the manufacturer's instructions. The absence of DNA from the RNA samples was tested by null PCR amplification of the *psbA* gene in *E. salsugineum*, or *lhcb5* in *A. thaliana* (primers in Supplementary Table S2). cDNA synthesis was performed with the Tetro cDNA Synthesis Kit (Meridian Bioscience, OH, USA), using Oligo (dT)[18] primers, following manufacturer's instructions. qPCR reaction was performed using FastStart Universal SYBR-Green Master Mix (Rox, Roche, Switzerland), in a StepOnePlus™ Real-Time PCR System (Thermo Fisher Scientific, MA, USA). Cycling parameters were an initial step of 95 °C for 10 min and a two-step cycle of 95 °C for 15 s and 60 °C for 1 min, repeated 40 times. This was followed by the dissociation protocol to check for specific amplification and possible contamination. Gene quantification was determined based on the expression of the *PTOX* gene relative to the housekeeping gene *GADPH*[20]. For comparative purposes, the average threshold cycle values of the control samples (Day 0) for *E. salsugineum*, or the EsPTOX

background expression in the wt Arabidopsis Col-0, were used as reference, respectively. Three biological replicates, and three technical replicates were performed for each sample and gene. Data analysis was performed using the StepOne Software v2.3 (Thermo Fisher Scientific, USA). The primer pairs used are given in Supplementary Table S2.

## Time-course RNAseq analysis

Total RNA of *E. salsugineum* leaves were extracted from plants subjected to high light stress (HL, 800 µmol m$^{-2}$ s$^{-1}$) at days 0, 3, 5 and 10 ($n = 3$), following the manufacturer's instructions of the Plant Spectrum Total RNA Kit (Sigma, MO, USA). RNA was checked for quality and quantified as mentioned above. RNA (1 µg per sample) was delivered to the Genomic Technology Facility of The University of Manchester, for sequencing and analysis. Unmapped paired-end sequences from an Illumina HiSeq4000/NovaSeq 6000 sequencer were tested by FastQC (http://www.bioinformatics.babraham.ac.uk/projects/fastqc/).

Sequence adapters were removed, and reads were quality trimmed using Trimmomatic_0.39[48]. The reads were mapped against the reference genome of *E. salsugineum* v1.0 from (Esalsugineum_173_v1.0 https://data.jgi.doe.gov/refine-download/phytozome?organism= Esalsugineum&expanded=173)[49], and counts per gene were calculated using STAR_2.7.7a[50]. Approximately 90% of the reads generated were successfully mapped, of which more than 80% were assigned to genes and used for analysis (Supplementary Fig. S20A). Normalization, Principal Components Analysis, and differential expression (DE) were calculated with DESeq2_1.18.1[51]. PCA plot shows that most of the data variance can be explained by the HL sampling day and, together with a tight clustering of replicates, indicates that gene expression changes are the consequence of the progression of stress (Supplementary Fig. S20B). Adjusted *p*-values were corrected for multiple testing (Benjamini and Hochberg method).

A list of differentially expressed genes (4010 genes) was created by filtering for genes with a $p_{adj} < 0.01$ in any of the three DE tests (Day 3 vs. Day 0, Day 5 vs. Day3 and Day 10 vs. Day 5). These genes were segregated into 5 clusters based on similarity of expression profile across the dataset using a k-means clustering algorithm. Clustering was performed on the means of each sample group (log$_2$) that had been z-transformed (for each gene the mean set to zero, standard deviation to 1). K-means clustering was performed based on similarity of profiles (Hartigan–Wong algorithm) across the dataset using kmeans (R version 4.2.0).

Gene ontology (GO) enrichment analysis was performed on each of the gene clusters using the best *A. thaliana* homolog (https:// phytozome.jgi.doe.gov/pz/portal.html; last accessed on 01 October 2022), and clusterProfiler v4.4.4[52]. RNAseq validation was performed by measuring the expression changes of 3 independent genes, which were classified into different expression clusters along the high light treatment (*Thhalv10025624m* –PTOX-, *Thhalv10022631m* and *Thhalv10026385m*), by qPCR. A linear regression analysis of the values of RNA-Seq and qPCR expression was performed, resulting in a R$^2$ = 0.91 (Supplementary Fig. S21).

## SDS/PAGE and immunoblot analysis

Leaf proteins were extracted using extraction buffer (50 mM Tris-HCl pH 7.8) containing 1 mM EDTA, 2% Triton X-100, 10% glycerol, and 0.5 mM of phenylmethylsulfonyl fluoride (PMSF). Aliquots containing 60 µg of protein from a given supernatant preparation were mixed with 4x Laemmli Sample buffer (BIO-RAD, CA, USA), heated at 95 °C for 5 min and separated by electrophoresis on a Mini-PROTEAN Any kD TGX Stain-Free Gel (BIO-RAD, USA) under denaturing conditions. For immunoblot analysis, proteins were transferred to a nitrocellulose blotting membrane by semidry electroblotting, in a Trans-Blot Turbo Transfer System (BIO-RAD, USA). Membranes were blocked overnight at 4 °C, using Western Blocker solution (Sigma, MO, USA). Then, for immunodetection, membranes were incubated overnight at 4 °C with

PTOX antibody (1:10,000 dilution, AS16 3692, Agrisera). Detection was performed with HRP-conjugated secondary antibody (1:25,000 anti-rabbit IgG, AS09 602, Agrisera) and ECL detection reagents (Thermo Fisher Scientific, MA, USA), according to the manufacturer's instructions. Both Stain-Free and immunoblots were imaged using the Chemidoc MP system (BIO-RAD, USA).

## Chlorophyll measurements

Leaves were ground with liquid nitrogen, weighed fresh and chlorophyll extracted using 80% (v/v) acetone. Chlorophyll content (nmol.mg FW$^{-1}$) was calculated according to Porra et al.[53]. Briefly, the absorbance of the extract was measured at 646, 663 and 750 nm, and Chl *a*, Chl *b*, and Total Chl were calculated as followed: (1) Chl *a* (µg/mL) = 12.25 A$_{(663-750)}$ − 2.55 A$_{(646-750)}$, (2) Chl *b* (µg/mL) = 20.31 A$_{(663-750)}$ − 4.91 A$_{(646-750)}$ and (3) Total Chl (µg/mL) = 17.76 A$_{(663-750)}$ − 7.34 A$_{(646-750)}$.

## Photosynthetic parameters

Photosynthetic parameters were measured as described previously by Stepien and Johnson[4]. Chlorophyll *a* fluorescence emission was measured using a PAM-101 fluorometer (Heinz Walz, GmbH, Germany), being ΦPSII calculated as described by Maxwell and Johnson[54]. PSII relative electron transport rate (rETR) was calculated by multiplying ΦPSII by the corresponding actinic light intensity used to illuminate leaves[54]. Gas exchange parameters were measured using a LI-6400*XT* infrared gas analyzer (LI-COR, NE, USA). To measure net photosynthetic rates (A, µmol CO$_2$ m$^{-2}$ s$^{-1}$) under growth conditions (400 ppm CO$_2$ and 100 µmol m$^{-2}$ s$^{-1}$ light), leaves were illuminated for 30 min to achieve steady-state values. The maximum capacity for photosynthesis (P$_{max}$) was estimated under 2000 ppm CO$_2$ and 1000 µmol m$^{-2}$ s$^{-1}$ light, by subtracting dark respiration rates as previously described[55].

## Measurements of PTOX activity

PTOX activity was estimated by two independent methods. First, leaves were placed in the chamber of a CIRAS1 infrared gas analyzer (PP systems, MA, USA) to supply CO$_2$ saturating conditions (2000 ppm CO$_2$), and 21% or 1% O$_2$. Oxygen gas mixtures were supplied by mixing compressed oxygen and nitrogen from cylinders (BOC Gases, UK) using a MKS controller (MKS Instruments, MA, USA). Chlorophyll *a* fluorescence readings were taken by placing the optical fiber of the PAM-101 on top of the chamber of the IRGA, as in Essemine et al.[56]. Under saturating CO$_2$ conditions, photorespiration is assumed negligible and, thus, differences in electron transport between 21% and 1% O$_2$ can be consider as an estimation of PTOX activity[4,6,20]. To confirm activation of PTOX, detached leaves were vacuum infiltrated with either water or 1 mM nPG (n-propyl gallate, 3,4,5-trihydroxy-benzoic acid-*n*-propyl ester, Sigma), a known specific inhibitor of PTOX[16,20,57]. ΦPSII and PSII rETR were calculated as previously described, by illuminating the leaves with 1000 µmol.m$^{-2}$.s$^{-1}$ light for 30 min to achieve steady-state conditions. In both types of measurements, plants or detached leaves (respectively) were incubated in darkness for 20 min before the initiation of the experiment.

## Photoinhibition treatment

The photoinhibition treatment was performed as previously described by Rosso et al.[17]. Detached leaves were incubated at 4 °C and 1000 µmol m$^{-2}$ s$^{-1}$ light (photoinhibition treatment, St), or at room temperature and 10 µmol m$^{-2}$ s$^{-1}$ light (control, Ct) for 8 h. Photoinhibition was measured as the decrease in the maximum quantum yield of PSII (Fv/Fm), measured by a FluorPen FP 100 (PSI, Drásov, Czech Republic) every 2 h.

## ROS quantification by 2′,7′-dichlorofluorescein diacetate (DCF-DA)

The fluorescence dye 2′,7′-dichlorofluorescein diacetate (DCF-DA, Sigma) was used to quantify ROS generation[58]. Detached leaves were

treated under photoinhibitory or control conditions (as previously described) for 2 h, and vacuum infiltrated with 10 mM Tris-HCl (pH 7.5) buffer with 50 µM DCF-DA. Leaves were incubated in darkness for 30 min, before illumination at 1000 µmol m$^{-2}$ s$^{-1}$ light and room temperature conditions, for 10 min. The aim of the stress treatment was to trigger PSII photodamage, while the role of the high light exposure after DCF-DA infiltration was to trigger ROS generation in the photoinhibited leaves. As fluorescence controls, leaves were incubated under the same conditions described above, but infiltrated with buffer solution without DCF-DA. ROS generation was assessed by analyzing the fluorescence of the leaves at 485 nm ($\lambda_{exc}$ DCF-DA, $\lambda_{em}$ = 535 nm) and 488 nm ($\lambda_{exc}$ chlorophyll autofluorescence, $\lambda_{em}$ = 670–730 nm) using a modular stereo fluorescence microscope Leica MZ 10 F and a Leica DFC7000 T camera (Leica Microsystems CMS GmbH, Germany). The fluorescence of each leave was quantified using the ImageJ software (National Health Institute, MD, USA). ROS subcellular generation was studied through confocal microscopy analysis in a Leica SP8 Upright confocal microscope (Leica Microsystems CMS GmbH, Germany). Pre-treated leaves were cut in 1 mm strips and excited at the wavelengths previously mentioned. Images were processed using the Leica LAS AF LITE software (Leica Microsystems CMS GmbH, Germany).

## Transmission electron microscopy (TEM)

Chloroplast ultrastructural analysis was performed in the Electron Microscopy Core Facility of the University of Manchester (Faculty of Biology, Medicine and Health, FBMH), using a Talos L120C TEM (ThermoFisher Scientific, MA, USA) at 120 kV accelerating voltage with Ceta CMOS camera. Leaf pieces (2 mm$^2$) were infiltrated and fixed with 2.5% glutaraldehyde and 4% paraformaldehyde, in 0.1 M HEPES buffer (pH 7.2), for 2 h at room temperature. Then, samples were washed three times with distilled water to remove the excess of fixative, and secondary fixed with 1% OsO$_4$ and 1.5% KFe(CN)$_6$ for 1 h. Samples were again washed three times with distilled water and incubated overnight with 1% uranyl acetate at 4 °C. Dehydration was performed with alcohols, for their inclusion in a low viscosity resin TAAB LV. Ultra-thin sections (70 nm) were cut and mounted on grids for TEM imaging. Image magnification for chloroplast ultrastructure analysis was 17,500X. Chloroplast ultrastructure analysis was performed with the FIJI Image J software (National Health Institute, MD, USA). For quantification purposes, a grana stack was defined as three or more thylakoid membranes stacked together, as stated by Mazur et al.[33]. Grana margins swelling was calculated by relativizing the height of the grana margins, to the height of the central grana[33], for each treatment independently.

## Statistical analysis

Statistical analysis was carried out using Prism (GraphPad Software, CA, USA). One-way ANOVA was performed, followed by Tukey test for multivariate analysis ($p < 0.05$). The Shapiro–Wilk test was used to test for normality ($p < 0.05$). For pair comparisons, a Student's $t$ test was performed with the same software ($p < 0.05$, two-sided). If treatments did not show equal variances, a Welch's $t$ test was performed instead ($p < 0.05$, two-sided).

## Reporting summary

Further information on research design is available in the Nature Portfolio Reporting Summary linked to this article.

## Data availability

Illumina reads generated during the Time-Course RNAseq of *Eutrema salsugineum* plants acclimated to high light are available in BioStudies of the EMBL-EBI [https://www.ebi.ac.uk/biostudies/], under the accession number E-MTAB-12913. Original electron microscopy and confocal images are available in the Zenodo database under the following https://doi.org/10.5281/zenodo.10230466. All other data generated or analyzed during this study are included in this published article (and its supplementary information files). Source data are provided with this paper.

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

## Acknowledgements

The authors thank Dr. Beata Czajkowska for technical support in the generation of the transgenic *Arabidopsis thaliana* Es-PTOX lines, Mr. Josef Olivier for useful discussions and Dr. Minsung Kim for critical reading of the manuscript. The authors thank the staff in the EM Core Facility in the Faculty of Biology, Medicine and Health (University of Manchester) for their assistance, and the Wellcome Trust for equipment grant support to the EM Core Facility (SCR_021147). We further acknowledge Dr. Aleksandr Mironov for his assistance in the TEM study. The authors also would like to thank Dr. Leo Zeef from the Bioinformatics Core Facility of the University of Manchester, for assistance in the RNAseq analysis. This work was supported by BBSRC research grant BB/J04103/1, awarded to G.N.J and P.G.

## Author contributions

P.C. performed, designed, and analyzed the data of all experiments. J.S. participated in some physiological experiments and in the confocal microscopy analysis. P.G. and G.J. conceived the project. G.J. designed and supervised the experiments and analyzed the data. The article was written by P.C., P.G. and G.J. All authors revised and approved the final version of the manuscript.

## Competing interests

The authors declare no competing interests.
