## [Peer Review File · Nature Communications]

REVIEWER COMMENTS

Reviewer #1 (Remarks to the Author):

The authors of this manuscript investigated how the activity of plastid terminal oxidase (PTOX) is induced to achieve a photoprotective function in land plants *Eutrema salsugineum* and *Arabidopsis thaliana*. Maximum photosynthetic capacity, quantum yield of photosystem II, electron transport rate of photosystem II, photosynthetic assimilation rates, and Fv/Fm were examined for wild-type and mutant plants under different light intensities. Chloroplast ultrastructure analysis was performed by the transmission electron microscopy. High-light-induced changes in gene expression in *Eutrema salsugineum* were also examined. The authors concluded that unstacking of thylakoids enables PTOX to be in close proximity to photosystem II, resulting in an effective electron transfer between them to exhibit a photoprotective function. Experiments and analyses appear to be systematically carried out. The followings are this reviewer's questions and suggestions to improve the manuscript.

To what extent do thylakoids need to be separated for PTOX to be active? In this sense, how did the authors distinguish between "stacking" and "non-stacking"?

In line 156, it would be better to clearly describe what changes are caused in thylakoid stacking. Do the authors mean a decrease in stacking?

In Figure 3, chloroplast ultrastructure parameters are given. Besides the numbers of chloroplasts and grana stacks, please describe in the figure caption how many cells, leaves, and plants were examined.

Please explain what Figure 3H and 3I depict in the caption.

In Figure 4, the colors of leaves are different in A, E, I, and N. Please explain the differences in pigment contents among these four, and how the differences affect the photosynthetic activity.

Raw data for "Fv/Fm" need to be provided.

Typos need to be fixed. For example, "simply" is doubled in lines 110–111, two dots are not necessary in line 211, and so on.

Reviewer #2 (Remarks to the Author):

This study is about the thylakoid-associated enzyme called plastid terminal oxidase (PTOX) which catalyzes the electron transport from plastoquinol (generated mainly by photosystem II) to molecular oxygen. This electron bypass could be photoprotective under for example high light stress where photosystem II might generate too much reducing power which could lead to the generation of toxic reactive oxygen species. The study finds a correlation between ultrastructural thylakoid changes and PTOX activity. This link between thylakoid ultrastructure and PTOX activity is interesting and novel. It provides evidence for a PSII-PTOX proximity model proposing that both protein complexes must be in close contact to allow electron transport from PSII to PTOX via plastoquinol. It is hypothesized that PTOX is activated by unstacking of grana. The electron transport to PTOX can act as a safety valve for excess electrons, which accumulate under stress. Although interesting, the manuscript has several flaws detailed below. A main concern is what exactly is meant by 'unstacking'. Furthermore, it should be addressed how thylakoid unstacking activates electron transfer from PSII to PTOX, i.e., why does a close contact between the two proteins activate electron transport from plastoquinol to PTOX?

1. Grana stacking (lines 151, 152): It is unclear how the authors conclude that the level of stacking is reduced after HL treatment in *Eutrema*. The results show a HL triggered reduction in grana height and number of membranes per grana stack. But at the same time an increase in grana diameter is seen. The latter will lead to an increase in the stacked area whereas the former reduces stacking. It is not obvious from the data presented why the level of stacking is reduced after HL. In this context it should be precisely defined what stacking means. In general, unstacking analysis is not precise. For example, lines 160-162: A decrease in the numbers of thylakoids per grana and total height does not automatically mean unstacking. It was reported that also the number of grana per chloroplast can change under similar conditions, i.e., unstacking on the single grana level can be compensated by more grana stacks per chloroplast. Therefore, a more thorough stacking analysis is advisable.

2. L 156: It is published that HL leads to preferential bulging of the peripheral parts of grana leading the core of the grana unaffected. This observation is neglected but seems highly relevant to this story claiming that HL causes unstacking and thereby allows closer contact between PSII and PTOX. Also, it would be worthwhile to check whether this bulging is apparent in the current electron micrographs. It is not possible to evaluate this from the images provided.

3. To verify that under 2000 ppm CO₂ comparing 21% with 1% O₂ estimates PTOX activity the PTOX inhibitor nPG was vacuum infiltrated into leaves (Fig. 1D). Authors made the case that nPG-triggered decrease in Phi₂ is similar under 21% O₂ and 1% O₂. If this is a specific effect of nPG on PTOX this conclusion makes sense. However, nPG leads to a decline of Phi₂ under all tested conditions. Thus, an unspecific nPG inhibition of Phi₂ that is independent of PTOX, cannot be ruled out. Since this is a critical experiment, a test that nPG under the given experimental conditions specifically inhibits PTOX would be critical. This independent test could include nPG tests with PTOX mutants, for example.

4. PTOX activity was measured under 2000 ppm CO₂ to minimize photorespiration. How safe is it to assume that in *Eutrema* under 2000 ppm CO₂, photorespiration is apparently non existing?

5. PTOX protein levels: transcript levels are high at 3 and 5 days of HL. PTOX Western blots are shown for 10 days only (fig. 1E). It would be interesting to see whether PTOX protein is elevated at days 3 and 5 when the transcript levels are increased and not at day 10 when the PTOX transcript level is low again.

Along these lines the statement in lines 110,111: "Therefore, the activity of PTOX in the HL-treated plants cannot simply be explained simply by an increase in relative PTOX protein concentration." should be treated more carefully. Assuming that the PTOX protein level is higher at days 3 and 5 (see above for additional WBs) it could well be that this higher content can explain higher PTOX activity around these days. Finally, the loading control in fig. 1F is not convincing. It seems that the HL samples contain significantly less overall protein. This should be checked.

6. PSII ETR: Line 401: "PSII ETR was calculated by multiplying Φ PSII by the corresponding actinic light intensity used to illuminate leaves⁵¹" This is not correct. PSII ETR should be calculated by this equation:
$$\text{PSII ETR} = \text{PAR} * \Phi\text{PSII} * n * \text{abs}$$

With n, antenna size ratio PSII to PSI (typically $n = 0.5$) and abs, leaf absorption.

This raises the question whether it is valid to assume that n is constant, in particular under 21% and 1% O₂. Can it be ruled out that different O₂ concentrations lead to changes in PSII to PSI antenna size ratio?

7. For transmission, electron microscopy leaf pieces were fixed for 2 hours at room temperature with glutaraldehyde. Were light-treated samples still illuminated during the fixation process? If not, how safe is it to assume that the light-adapted state was preserved during the 2 hours fixation? If yes, does the combination of glutaraldehyde (act as uncoupler) and high light causes artificial alterations of the thylakoid ultrastructure?

8. Figures 3F and 3G: Labels are missing. What is Ct and what is HL?

REVIEWER COMMENTS

Reviewer #1 (Remarks to the Author):

The authors of this manuscript investigated how the activity of plastid terminal oxidase (PTOX) is induced to achieve a photoprotective function in land plants *Eutrema salsugineum* and *Arabidopsis thaliana*. Maximum photosynthetic capacity, quantum yield of photosystem II, electron transport rate of photosystem II, photosynthetic assimilation rates, and Fv/Fm were examined for wild-type and mutant plants under different light intensities. Chloroplast ultrastructure analysis was performed by the transmission electron microscopy. High-light-induced changes in gene expression in *Eutrema salsugineum* were also examined. The authors concluded that unstacking of thylakoids enables PTOX to be in close proximity to photosystem II, resulting in an effective electron transfer between them to exhibit a photoprotective function. Experiments and analyses appear to be systematically carried out. The followings are this reviewer's questions and suggestions to improve the manuscript.

To what extent do thylakoids need to be separated for PTOX to be active? In this sense, how did the authors distinguish between "stacking" and "non-stacking"?

We appreciate the comment regarding to what extent thylakoids need to be separated for PTOX activation. This is an interesting point to discuss and further investigate. Our results clearly demonstrate that unstacking (definition below) allows PTOX to be a sink of electrons of PSII, with this electron transfer being photoprotective. Please see also response to Referee 2, where we added additional data about thylakoid grana modifications.

We defined a grana stack as a structure having at least three layers of thylakoid membranes stacked together (please see L479). This criterion was previously used in different studies in which they analysed the correlation between physiological processes in plants and chloroplast ultrastructure modifications (Chuartzmann et al. 2008; Schumann et al. 2007; Wood et al. 2018).

The need for a precise definition of grana stacking, and its implications in ultrastructure analysis, has been previously discussed by Mazur et al. (2021). In this article, authors stated that the lowest grana stack that could be defined has to be built at least of three thylakoid layers. We now cited this article in the Materials and Methods sections (L480), and we also included a sentence in the results section about how we defined a grana stack for clarification (please see L152).

References:

Chuartzman, S. G., Nevo, R., Shimoni, E., Charuvi, D., Kiss, V., Ohad, I., ... & Reich, Z. (2008). Thylakoid membrane remodeling during state transitions in *Arabidopsis*. *The Plant Cell*, 20(4), 1029-1039.

Mazur, R., Mostowska, A., & Kowalewska, Ł. (2021). How to measure grana-ultrastructural features of thylakoid membranes of plant chloroplasts. *Frontiers in Plant Science*, 12, 756009.

Schumann, T., Paul, S., Melzer, M., Dörmann, P., & Jahns, P. (2017). Plant growth under natural light conditions provides highly flexible short-term acclimation properties toward high light stress. *Frontiers in Plant Science*, 8, 681.

Wood, W. H., MacGregor-Chatwin, C., Barnett, S. F., Mayneord, G. E., Huang, X., Hobbs, J. K., ... & Johnson, M. P. (2018). Dynamic thylakoid stacking regulates the balance between linear and cyclic photosynthetic electron transfer. *Nature plants*, 4(2), 116-127.\

In line 156, it would be better to clearly describe what changes are caused in thylakoid stacking. Do the authors mean a decrease in stacking?

Yes, this is what we intended. A decrease in grana stacking, meaning fewer thylakoid layers per grana stack, has been correlated with growth at high light intensities in several species, such as *Arabidopsis*, soybean and radish (Ballantine et al 1970; Lichtenthaler et al. 1983; Schumann et al. 2017; Flannery et al. 2021). To clarify this point, the following change was made in the manuscript:

L164. 'HL is known to cause a decrease in grana stacking in many plant species^{31,32}, suggesting that this may give rise to PTOX activity'.

References:

Ballantine, J. E. M., & Forde, B. J. (1970). The effect of light intensity and temperature on plant growth and chloroplast ultrastructure in soybean. *American Journal of Botany*, 57(10), 1150-1159.

Flannery, S. E., Hepworth, C., Wood, W. H., Pastorelli, F., Hunter, C. N., Dickman, M. J., ... & Johnson, M. P. (2021). Developmental acclimation of the thylakoid proteome to light intensity in *Arabidopsis*. *The Plant Journal*, 105(1), 223-244.

Lichtenthaler, H. K., Burgstahler, R., Buschmann, C., Meier, D., Prenzel, U., & Schönthal, A. (1983). Effect of high light and high light stress on composition, function and structure of the photosynthetic apparatus. In *Effects of Stress on Photosynthesis: Proceedings of a conference held at the 'Limburgs Universitair Centrum' Diepenbeek, Belgium, 22–27 August 1982* (pp. 353-370). Springer Netherlands.

Schumann, T., Paul, S., Melzer, M., Dörmann, P., & Jahns, P. (2017). Plant growth under natural light conditions provides highly flexible short-term acclimation properties toward high light stress. *Frontiers in Plant Science*, 8, 681.

In Figure 3, chloroplast ultrastructure parameters are given. Besides the numbers of chloroplasts and grana stacks, please describe in the figure caption how many cells, leaves, and plants were examined.

We now included the information requested in the legend of Figure 3 as follows:

'A minimum of 25 non-overlapping fields of view of the ultra-thin sections were examined, and leaf pieces from two plants were fixed for the analysis.'

Please explain what Figure 3H and 3I depict in the caption.

We apologize for the omission. The following sentences were added to the Figure legend.

'H. ΦPSII was measured at 1,000 μmol m⁻² s⁻¹ light and 2,000 μl.l⁻¹ CO₂, and at 21% or 1% O₂, in *Arabidopsis* plants exposed to 12 d Ct or HL conditions. I. ΦPSII was measured in vacuum-infiltrated

leaves of 12 d Ct or HL-treated Arabidopsis plants, at 1,000 $\mu\text{mol m}^{-2} \text{s}^{-1}$ light and 2,000 $\mu\text{l.l}^{-1}$ CO_2 . Infiltration was performed with water or 1mM nPG.'

In Figure 4, the colors of leaves are different in A, E, I, and N. Please explain the differences in pigment contents among these four, and how the differences affect the photosynthetic activity.

As stated on L186 of the manuscript, the *chl1-3* genetic backgrounds are chlorophyll *b*-less mutants, which do not produce chlorophyll *b* (chlorophyll concentration measurements are given in Supp Fig. 19). This is the main reason why the *chl1-3* and the *chl1-3xlhcb5* genotypes show a pale green color when compared to the wt background.

The physiological analysis of the *chl1-3* and *chl1-3xlhcb5* genotypes was carried out by Kim et al. (2009), in a paper where they study the effect of these mutations in photosynthetic activity. The authors described that the mutations in *chl1-3* and *chl1-3xlhcb5* are associated with a loss of photochemical efficiency of PSII and Pmax, due to their thylakoid stacking impairment. Moreover, a higher susceptibility to photoinactivation of PSII was found in the mutants, and effect which was more pronounced in the *chl1-3xlhcb5* genotype due to the lack of LHCB5. We have added a sentence about this to the manuscript (L189) though unfortunately we are limited for space to explain further.

The photosynthetic phenotype described by Kim et al. (2009) was also confirmed in our experiments, observing a decrease in ΦPSII , Pmax and Fv/Fm in the *chl1-3* and *chl1-3xlhcb5* mutants, when compared to the wt genotype. This effect was also more pronounced in the double mutant background (please see Fig. 4C, G, K and P for PSII rETR at 1 % O_2 ; Fig. 5 G-I for Pmax, and Fig. 6A-C for Fv/Fm). Importantly, these alterations in photosynthetic activity do not affect our conclusions and statements regarding the mechanism of activation of PTOX. In our study, all the comparisons were performed between genetic backgrounds over-expressing or not PTOX, and comparisons between genotypes were avoided.

Reference:

Kim, E. H., Li, X. P., Razeghifard, R., Anderson, J. M., Niyogi, K. K., Pogson, B. J., & Chow, W. S. (2009). The multiple roles of light-harvesting chlorophyll a/b-protein complexes define structure and optimize function of Arabidopsis chloroplasts: a study using two chlorophyll b-less mutants. *Biochimica et Biophysica Acta (BBA)-Bioenergetics*, 1787(8), 973-984.

Raw data for "Fv/Fm" need to be provided.

We have included the raw data for Fv/Fm in the raw data spreadsheet, which is uploaded with the resubmission.

Typos need to be fixed. For example, "simply" is doubled in lines 110–111, two dots are not necessary in line 211, and so on.

We thank reviewer for the comment. We have carefully proof-read the manuscript and hope we have corrected all these.

Reviewer #2 (Remarks to the Author):

This study is about the thylakoid-associated enzyme called plastid terminal oxidase (PTOX) which catalyzes the electron transport from plastoquinol (generated mainly by photosystem II) to molecular oxygen. This electron bypass could be photoprotective under for example high light stress where photosystem II might generate too much reducing power which could lead to the generation of toxic reactive oxygen species. The study finds a correlation between ultrastructural thylakoid changes and PTOX activity. This link between thylakoid ultrastructure and PTOX activity is interesting and novel. It provides evidence for a PSII-PTOX proximity model proposing that both protein complexes must be in close contact to allow electron transport from PSII to PTOX via plastoquinol. It is hypothesized that PTOX is activated by unstacking of grana. The electron transport to PTOX can act as a safety valve for excess electrons, which accumulate under stress. Although interesting, the manuscript has several flaws detailed below. A main concern is what exactly is meant by 'unstacking'. Furthermore, it should be addressed how thylakoid unstacking activates electron transfer from PSII to PTOX, i.e., why does a close contact between the two proteins activate electron transport from plastoquinol to PTOX?

First, we would like to thank the Reviewer's for the comments and suggestions. Regarding the concern related to what do we mean by 'unstacking', this is addressed in the reply to Reviewer 1.

The need for a precise definition of grana stacking, and its implications for ultrastructure analysis, has been previously discussed by Mazur et al. (2021). In this article, authors already stated that the lowest grana stack that could be defined has to be built at least of three thylakoid layers. We now cited this article in the Materials and Methods sections, and we also included a sentence in the results section about how we defined a grana stack for clarification (please see L480 and L152, respectively).

In relation to the second point raised by the Reviewer, as is stated in the Introduction (L73), PSII and PTOX are located in different sub-compartments of the chloroplast. While PSII is specifically targeted to the grana (Rantala et al. 2020), PTOX has been shown to be localized to the stromal lamellae (Lennon et al. 2003). As it is bound to and protrudes from the surface of the membrane, it is likely to be excluded from the inter-membrane spaces in the thylakoid stacks. Since PTOX oxidizes plastoquinol and reduces O₂ to produce H₂O, its activity will depend on the availability of its substrates. It has been shown that PQ diffusion is highly restricted in the thylakoid membranes of plants (Kirchoff, 2014), and particularly to small diffusion areas around PSII (Kirchhoff et al. 2000; Lavergne et al. 1992). Thus, based on the PQ pool diffusion rates, the distance between PSII and PTOX would preclude a significant electron flow between these two complexes. PTOX would need to be located close to PSII to function as an efficient electron sink.

When the thylakoids are unstacked, the relative area of contact between grana and stromal lamellae membranes is increased by the smaller grana size, and a widely distribution of grana and stroma lamellae complexes are observed in the grana margins (Wood et al. 2018). Evidence also shows that unstacking enhances protein mobility in the thylakoid membranes due to lower protein-packing densities (Kirchhoff et al. 2008; 2013). This increases the probability of PTOX getting closer to a reduced PQ pool associated with PSII, and as such, its capability for acting as an electron sink pathway of the latter. Moreover, it has been demonstrated that unstacking facilitates the diffusion of the PQ pool,

promoting a more efficient electron flow between the complexes in the grana and the stroma lamellae (Wood et al. 2018). Consequently, unstacking not only increases the probability of mixture between PSII and PTOX, but also facilitates electron transport through diffusion of the reduced PQ pool associated with the photosystem.

References:

- Chuartzman, S. G., Nevo, R., Shimoni, E., Charuvi, D., Kiss, V., Ohad, I., ... & Reich, Z. (2008). Thylakoid membrane remodeling during state transitions in Arabidopsis. *The Plant Cell*, 20(4), 1029-1039.
- Kirchhoff, H., Horstmann, S., & Weis, E. (2000). Control of the photosynthetic electron transport by PQ diffusion microdomains in thylakoids of higher plants. *Biochimica et Biophysica Acta (BBA)-Bioenergetics*, 1459(1), 148-168.
- Kirchhoff, H. (2014). Diffusion of molecules and macromolecules in thylakoid membranes. *Biochimica et Biophysica Acta (BBA)-Bioenergetics*, 1837(4), 495-502.
- Kirchhoff, H., Sharpe, R. M., Herbstova, M., Yarbrough, R., & Edwards, G. E. (2013). Differential mobility of pigment-protein complexes in granal and agranal thylakoid membranes of C3 and C4 plants. *Plant physiology*, 161(1), 497-507.
- Kirchhoff, H. (2008). Molecular crowding and order in photosynthetic membranes. *Trends in plant science*, 13(5), 201-207.
- Mazur, R., Mostowska, A., & Kowalewska, Ł. (2021). How to measure grana-ultrastructural features of thylakoid membranes of plant chloroplasts. *Frontiers in Plant Science*, 12, 756009.
- Lavergne, J., Bouchaud, J. P., & Joliot, P. (1992). Plastoquinone compartmentation in chloroplasts. II. Theoretical aspects. *Biochimica et Biophysica Acta (BBA)-Bioenergetics*, 1101(1), 13-22.
- Lennon, A. M., Prommeenate, P. & Nixon, P. J. Location, expression and orientation of the putative chlororespiratory enzymes, Ndh and IMMUTANS, in higher-plant plastids. *Planta* 218, 254–260 (2003).
- Rantala, M., Rantala, S. & Aro, E.-M. Composition, phosphorylation and dynamic organization of photosynthetic protein complexes in plant thylakoid membrane. *Photochem. Photobiol. Sci.* (2020). doi:10.1039/d0pp00025f
- Schumann, T., Paul, S., Melzer, M., Dörmann, P., & Jahns, P. (2017). Plant growth under natural light conditions provides highly flexible short-term acclimation properties toward high light stress. *Frontiers in Plant Science*, 8, 681.
- Wood, W. H., MacGregor-Chatwin, C., Barnett, S. F., Mayneord, G. E., Huang, X., Hobbs, J. K., ... & Johnson, M. P. (2018). Dynamic thylakoid stacking regulates the balance between linear and cyclic photosynthetic electron transfer. *Nature plants*, 4(2), 116-127.

1. Grana stacking (lines 151, 152): It is unclear how the authors conclude that the level of stacking is reduced after HL treatment in *Eutrema*. The results show a HL triggered reduction in grana height and number of membranes per grana stack. But at the same time an increase in grana diameter is seen. The latter will lead to an increase in the stacked area whereas the former reduces stacking. It is not obvious from the data presented why the level of stacking is reduced after HL. In this context it should be precisely defined what stacking means. In general, unstacking analysis is not precise. For example, lines 160-162: A decrease in the numbers of thylakoids per grana and total height does not automatically mean unstacking. It was reported that also the number of grana per chloroplast can change under

similar conditions, i.e., unstacking on the single grana level can be compensated by more grana stacks per chloroplast. Therefore, a more thorough stacking analysis is advisable.

As commented above, we defined as a grana stack at least three layers of thylakoid membranes stacked together (Mazur et al. 2021). Based on this definition, we considered the degree of stacking/unstacking as depending on the number of layers which build a grana stack in each treatment. This criterion, and the ultrastructure parameters calculated, were also used in different studies in which they analysed the correlation between physiological processes in plants and chloroplast ultrastructure modifications (Chuartzmann et al. 2008; Schumann et al. 2007; Armbruster et al. 2013; Yokoyama et al. 2016; Wood et al. 2018).

However, based on Reviewer's suggestion, we further analysed our TEM micrographs by calculating the total length of thylakoid membranes enclosed in grana stacks, per chloroplast. We assumed the length measured in sections to be a proxy of area.

The total length of the appressed thylakoid membranes within grana stacks, was significantly reduced in high light grown leaves in both *Eutrema* and *Arabidopsis* plants (please see Figure below). Interestingly, the length of stacked membranes was lower in the former species, in which a higher PTOX activity was observed under HL (please see Fig. 1B-D and Fig. 3H-I for comparison between PTOX activity in *Eutrema* and *Arabidopsis*, respectively). We have included these new data as a supplementary figure (Fig. S8), and the following sentence was included in the manuscript (added text underlined).

Results. L170:

‘Thus, thylakoid unstacking is also associated with PTOX activity under HL in Arabidopsis. However, the structural changes in Arabidopsis contrasted with those in Eutrema. Arabidopsis had more grana stacks under LL conditions than Eutrema, and that number increased further under HL (Fig. 3J c.f. 3B). At the same time, granal diameter decreased at HL in Arabidopsis, whilst increasing in Eutrema (Fig. 3M c.f. 3E). Nevertheless, the total length of thylakoid membranes appressed within grana stacks was significantly reduced by HL, both in Eutrema and Arabidopsis plants (Fig. S8).’

Figure S8. Measurements of the total length of thylakoid membranes appressed within grana stacks. The total length (µm) was measured in Eutrema (A) and Arabidopsis (B) wt plants. Ct, Control treatment; HL, High light-treated plants. Measurements were performed after 10 d of HL in Eutrema and 12 d of HL in Arabidopsis.). A minimum of 10 non-overlapping fields of view of the ultra-thin sections were examined, and leaf pieces of two plants were fixed for the analysis. The asterisk indicates differences between Ct and HL treatments (Student’s test, **** $p < 0.0001$).

References:

- Mazur, R., Mostowska, A., & Kowalewska, Ł. (2021). How to measure grana-ultrastructural features of thylakoid membranes of plant chloroplasts. *Frontiers in Plant Science*, 12, 756009.
- Schumann, T., Paul, S., Melzer, M., Dörmann, P., & Jahns, P. (2017). Plant growth under natural light conditions provides highly flexible short-term acclimation properties toward high light stress. *Frontiers in Plant Science*, 8, 681.
- Chuartzman, S. G., Nevo, R., Shimoni, E., Charuvi, D., Kiss, V., Ohad, I., ... & Reich, Z. (2008). Thylakoid membrane remodeling during state transitions in Arabidopsis. *The Plant Cell*, 20(4), 1029-1039.
- Wood, W. H., MacGregor-Chatwin, C., Barnett, S. F., Mayneord, G. E., Huang, X., Hobbs, J. K., ... & Johnson, M. P. (2018). Dynamic thylakoid stacking regulates the balance between linear and cyclic photosynthetic electron transfer. *Nature plants*, 4(2), 116-127.

Yokoyama, R., Yamamoto, H., Kondo, M., Takeda, S., Ifuku, K., Fukao, Y., ... & Shikanai, T. (2016). Grana-localized proteins, RIQ1 and RIQ2, affect the organization of light-harvesting complex II and grana stacking in Arabidopsis. *The Plant Cell*, 28(9), 2261-2275.

Armbruster, U., Labs, M., Pribil, M., Viola, S., Xu, W., Scharfenberg, M., ... & Leister, D. (2013). Arabidopsis CURVATURE THYLAKOID1 proteins modify thylakoid architecture by inducing membrane curvature. *The Plant Cell*, 25(7), 2661-2678.

2. L 156: It is published that HL leads to preferential bulging of the peripheral parts of grana leading the core of the grana unaffected. This observation is neglected but seems highly relevant to this story claiming that HL causes unstacking and thereby allows closer contact between PSII and PTOX. Also, it would be worthwhile to check whether this bulging is apparent in the current electron micrographs. It is not possible to evaluate this from the images provided.

We agree with the reviewer's comment on the possible significance of alterations in grana margins leading to a closer contact between PSII and PTOX and have therefore examined this using existing images. As stated, high light has been reported to induce grana margin expansions and lateral destacking of the peripheral grana, which has been suggested to facilitate the PSII repair cycle (Herbstová et al. 2012; Puthiyaveetil et al. 2014; Yoshioka-Nishimura et al. 2014; Ruban and Johnson, 2015). This phenomenon could also potentially increase the accessibility of PTOX to PSII, and the diffusion of the reduced PQ pool associated to the photosystem. The swelling or bending of the grana marginal membranes was calculated as described by Mazur et al. (2021). Briefly, the height of the central and marginal regions of the grana stacks were measured, being the latter relativized to the former (Height of marginal grana/Height of central grana).

As shown in the Figure below, high light did not induce marginal swelling of the grana stacks under our experimental conditions, in either species studied. Thus, based on these results, we cannot relate swelling of the grana margins with PTOX activation process.

The data regarding marginal grana swelling was added as a supplementary figure (please see Supplementary Figure 9, figure below) and the following sentences were added to the manuscript.

Results:

L177. 'HL has also been reported to induce grana margin expansions and swelling of the peripheral grana, which has been suggested to facilitate the PSII repair cycle (Puthiyaveetil et al. 2014; Yoshioka-Nishimura et al. 2014). However, none of these phenomena were observed under our experimental conditions, in either of the species studied (Fig. S9).'

Materials and Methods

L480. 'Grana margins swelling was calculated by relativizing the height of the grana margins, to the height of the central grana (Mazur et al. 2021), for each treatment independently.'

Figure S9. Measurements of the swelling of the thylakoid grana margins under stress conditions. The height of the central and marginal regions of the thylakoid membranes were measured, and the ratio of the latter with the former calculated in *Eutrema* (A) and *Arabidopsis* (B) wt plants. Ct, Control treatment; HL, High light-treated plants. Measurements were performed after 10 d of HL in *Eutrema* and 12 d of HL in *Arabidopsis*. At least 100 measurements were performed in each condition. A minimum of 25 non-overlapping fields of view of the ultra-thin sections were examined, and leaf pieces of two plants were fixed for the analysis.

References:

Herbstová, M., Tietz, S., Kinzel, C., Turkina, M. V., & Kirchhoff, H. (2012). Architectural switch in plant photosynthetic membranes induced by light stress. *Proceedings of the National Academy of Sciences*, 109(49), 20130-20135.

Mazur, R., Mostowska, A., & Kowalewska, Ł. (2021). How to measure grana-ultrastructural features of thylakoid membranes of plant chloroplasts. *Frontiers in Plant Science*, 12, 756009.

Puthiyaveetil, S., Tsabari, O., Lowry, T., Lenhert, S., Lewis, R. R., Reich, Z., & Kirchhoff, H. (2014). Compartmentalization of the protein repair machinery in photosynthetic membranes. *Proceedings of the National Academy of Sciences*, 111(44), 15839-15844.

Ruban, A. V., & Johnson, M. P. (2015). Visualizing the dynamic structure of the plant photosynthetic membrane. *Nature plants*, 1(11), 1-9.

Yoshioka-Nishimura, M., Nanba, D., Takaki, T., Ohba, C., Tsumura, N., Morita, N., ... & Yamamoto, Y. (2014). Quality control of photosystem II: direct imaging of the changes in the thylakoid structure and distribution of FtsH proteases in spinach chloroplasts under light stress. *Plant and Cell Physiology*, 55(7), 1255-1265.

3. To verify that under 2000 ppm CO₂ comparing 21% with 1% O₂ estimates PTOX activity the PTOX inhibitor nPG was vacuum infiltrated into leaves (Fig. 1D). Authors made the case that nPG-triggered decrease in Phi2 is similar under 21% O₂ and 1% O₂. If this is a specific effect of nPG on PTOX this conclusion makes sense. However, nPG leads to a decline of Phi2 under all tested conditions. Thus, an unspecific nPG inhibition of Phi2 that is independent of PTOX, cannot be ruled out. Since this is a critical experiment, a test that nPG under the given experimental conditions specifically inhibits PTOX would be critical. This independent test could include nPG tests with PTOX mutants, for example.

The reviewer states that nPG results in a decrease in Φ PSII under all conditions. However, this does not imply an alternative target for nPG. In Fig 1D, there is a small but significant difference between control leaves infiltrated with water vs nPG, which might reflect a small secondary effect. Equally, it may reflect a low activity of PTOX under these growth conditions. There is no significant difference between Ct-1% O₂-water and Ct- 21% O₂-nPG. Given the small differences and the errors inherent in the measurement, this is at the limits of our experimental accuracy.

In other measurements, there is no evidence for an alternative effect of nPG. For instance, there are no significant differences in Φ PSII of control Arabidopsis wt leaves infiltrated with 1 mM nPG or water (please see Fig. 3I and Fig. 4D). Moreover, consistent with our conclusions that thylakoid unstacking triggering PTOX activation, while no significant differences were observed in Φ PSII between the water and nPG infiltration treatments in the Arabidopsis wt EsPTOX (please see Fig. 4H), differences were measured in the *chl1-3*-EsPTOX and *chl1-3xlhcb5*-EsPTOX genotypes under the same conditions (please see Fig. 4M and Q, respectively). The fact that we measured an nPG effect in the same treatments in which we observed oxygen sensitivity of Φ PSII, increase our confidence that what we are measuring is PTOX activity.

It is also worth mentioning that nPG has been previously used in the literature to estimate PTOX activity (Stepien and Johnson, 2009, 2018; Josse et al. 2003), and that the specific inhibition of PTOX by nPG was tested by expressing the heterologous protein in *Escherichia coli* (Josse et al. 2000; Fu et al. 2009). Thus, we believe that our experiments with nPG are solid enough, and allow us to conclude that a Φ PSII decreased due to infiltration with the chemical, can be used as an estimation of PTOX activity.

References:

Fu, A., Aluru, M., & Rodermeil, S. R. (2009). Conserved active site sequences in Arabidopsis plastid terminal oxidase (PTOX): in vitro and in planta mutagenesis studies. *Journal of Biological Chemistry*, 284(34), 22625-22632.

Josse, E. M., Simkin, A. J., Gaffé, J., Labouré, A. M., Kuntz, M., & Carol, P. (2000). A plastid terminal oxidase associated with carotenoid desaturation during chromoplast differentiation. *Plant physiology*, 123(4), 1427-1436.

Josse, E. M., Alcaraz, J. P., Labouré, A. M., & Kuntz, M. (2003). In vitro characterization of a plastid terminal oxidase (PTOX). *European Journal of Biochemistry*, 270(18), 3787-3794.

Stepien, P., & Johnson, G. N. (2009). Contrasting responses of photosynthesis to salt stress in the glycophyte Arabidopsis and the halophyte *Thellungiella*: role of the plastid terminal oxidase as an alternative electron sink. *Plant physiology*, 149(2), 1154-1165.

Stepien, P., & Johnson, G. N. (2018). Plastid terminal oxidase requires translocation to the grana stacks to act as a sink for electron transport. *Proceedings of the National Academy of Sciences*, 115(38), 9634-9639.

4. PTOX activity was measured under 2000 ppm CO₂ to minimize photorespiration. How safe is it to assume that in *Eutrema* under 2000 ppm CO₂, photorespiration is apparently non existing?

As previously published (Stepien and Johnson, 2009), measurements of A/Ci curves show that 2000 ppm CO₂ is sufficient to saturate CO₂ fixation in *Eutrema*, and to the same extent, both under control and salt stress conditions (i.e. with stomata closed). Thus, we are confident to say that photorespiration is minimal under these conditions. Nevertheless, we cannot totally exclude that electron transport to

oxygen occurs via routes other than PTOX, which is why we include experiments using n-propyl gallate, to evidence PTOX activity using an alternative assay. The reference to Stepień and Johnson (2009) was added in L106.

Reference:

Stepień, P., & Johnson, G. N. (2009). Contrasting responses of photosynthesis to salt stress in the glycophyte *Arabidopsis* and the halophyte *Thellungiella*: role of the plastid terminal oxidase as an alternative electron sink. *Plant physiology*, 149(2), 1154-1165.

5. PTOX protein levels: transcript levels are high at 3 and 5 days of HL. PTOX Western blots are shown for 10 days only (fig. 1E). It would be interesting to see whether PTOX protein is elevated at days 3 and 5 when the transcript levels are increased and not at day 10 when the PTOX transcript level is low again. Along these lines the statement in lines 110,111: "Therefore, the activity of PTOX in the HL-treated plants cannot simply be explained simply by an increase in relative PTOX protein concentration." should be treated more carefully. Assuming that the PTOX protein level is higher at days 3 and 5 (see above for additional WBs) it could well be that this higher content can explain higher PTOX activity around these days. Finally, the loading control in fig. 1F is not convincing. It seems that the HL samples contain significantly less overall protein. This should be checked.

In the light of this feedback, we repeated the WB and loading control of Day 10, as suggested by the reviewer. The new gel and blot were replaced in Fig. 1E (please see below). The new data confirmed our previous results, showing that there are no substantive differences in PTOX content between *Eutrema* plants exposed to control or high-light treatments for 10 days.

Fig. 1E. PTOX protein content in 10 d Ct and HL-treated plants (protein loaded = 100 µg).

Regarding the PTOX protein content at days 3 and 5, when designing the experiment, we decided to focus on the days of the HL acclimation where PTOX activity was significantly observed. This was from Day 7 of the high light treatment onwards (please see Fig. 1B). Since the activity at Day 10 was the highest observed, we decided to analyse PTOX protein content at this day. We did not observe a higher PTOX content in the HL treatment, where PTOX activity was measured and, consequently, we conclude that PTOX activity was not necessarily correlated with its protein content. Similar conclusions were obtained by studying the *Arabidopsis* wt OE-PTOX lines (please see Fig. 4A-H), where a higher PTOX content was not leading to PTOX activity. Thus, we hypothesized that a different mechanism of activation needed to be involved, not necessarily related with protein content.

Then, based on our protein data at Day 10, we decided to explore the transcript kinetics of PTOX, to see if there was any regulation of the gene expression during the HL acclimation process. Unexpectedly, we

observed that peak of expression at Day 3 and 5, suggesting that PTOX might have other roles at shorter terms of high light acclimation. Nevertheless, for the purposes of the present study, we focussed our analysis on the processes taking place at longer times (particularly day 10), where PTOX activity as an electron sink of PSII was clearly measured (please see Fig. 1-B-D).

To clarify the reasoning behind the experiments performed, we modify the manuscript and changed the order of Fig. 1E and F.

L109. 'PTOX protein content was estimated at Day 10 to address if the activity measured was correlated with its protein expression levels. No differences were observed in PTOX protein content between treatments (Fig. 1E). Therefore, the activity of PTOX in the HL-treated plants cannot simply be explained by an increase in its relative protein concentration. PTOX transcript levels were measured to study if high light acclimation triggered any regulation in gene expression (Fig. 1F). A significant increase in PTOX transcript was observed in the HL-treated plants at Days 3 and 5, where PTOX activity was not observed, suggesting that PTOX might have a different role at shorter times of HL acclimation.'

6. PSII ETR: Line 401: "PSII ETR was calculated by multiplying Φ PSII by the corresponding actinic light intensity used to illuminate leaves⁵¹" This is not correct. PSII ETR should be calculated by this equation:

$$\text{PSII ETR} = \text{PAR} * \Phi\text{PSII} * n * \text{abs}$$

With n, antenna size ratio PSII to PSI (typically n = 0.5) and abs, leaf absorption.

This raises the question whether it is valid to assume that n is constant, in particular under 21% and 1% O₂. Can it be ruled out that different O₂ concentrations lead to changes in PSII to PSI antenna size ratio?

The referee notes that the equation we have used for calculating ETR is not the same as the widely used one that they quote. They also themselves highlight the problem with that equation, that most researchers use it making assumptions about photosystem ratio and absorbance, without establishing these experimentally. For this reason, we prefer to use a simpler version, without these assumptions, as discussed by Maxwell and Johnson (2000). We should however have made this clear by referring to "relative ETR", as we have done in other papers using this parameter. We have corrected this in the revised manuscript and figures.

The referee asks whether we can rule out oxygen affecting the photosystem antenna ratio. There is evidence that anaerobic conditions give rise to state transitions in *Chlamydomonas*, driving cells into state 2 in the dark, however there is no evidence of a similar response in higher plants as far as we are aware. In any case, the measurements we performed were all at an irradiance that is expected to suppress state transitions. Thus, we cannot totally rule out a change in PSII antenna size affecting ETR, however there is no evidence to support this and further, as with other measurements we have presented, we have PTOX inhibitor measurements to support our conclusions.

Reference:

Maxwell, K., & Johnson, G. N. (2000). Chlorophyll fluorescence—a practical guide. *Journal of experimental botany*, 51(345), 659-668.

7. For transmission, electron microscopy leaf pieces were fixed for 2 hours at room temperature with glutaraldehyde. Were light-treated samples still illuminated during the fixation process? If not, how safe

is it to assume that the light-adapted state was preserved during the 2 hours fixation? If yes, does the combination of glutaraldehyde (act as uncoupler) and high light causes artificial alterations of the thylakoid ultrastructure?

High light-treated leaves were taken from the cabinet under the defined conditions and were fixed with the fixing solution for 2 hours at room temperature and under room light conditions. Thus, samples were not high light illuminated during the fixing process. Nevertheless, this does not affect the preservation of the light-adapted state, since the ultrastructure modifications that are under analysis are the consequence of acclimation processes that takes days to occur (Ballantine et al. 1970; Schumann et al. 2017; Flannery et al. 2021; Lichtenthaler et al. 1982; Boardman, 1977).

We are aware that there are reversible processes taking place within the minute-hours range (e.g. state transitions, LHClI phosphorylation), time in which the samples are being fixed during the fixing protocol, which can dynamically affect chloroplast ultrastructure (Wood et al. 2019; Chuartzman et al. 2008). However, the fact that the chloroplast ultrastructure changes are only observed after days of HL acclimation, indicates that we are observing processes related to acclimation, and not to short-term light responses. This can be clearly seen in Figure 3A, where is evident that the ultrastructure changes in *Eutrema salsugineum* chloroplast exposed to HL appears mainly after 5 days of treatment, and are not present under control conditions or earlier times of stress.

References:

Ballantine, J. E. M., & Forde, B. J. (1970). The effect of light intensity and temperature on plant growth and chloroplast ultrastructure in soybean. *American Journal of Botany*, 57(10), 1150-1159.

Boardman, N. K. t. Comparative photosynthesis of sun and shade plants. *Annu. Rev. Plant Physiol.* 28, 355–377 (1977).

Chuartzman, S. G., Nevo, R., Shimoni, E., Charuvi, D., Kiss, V., Ohad, I., ... & Reich, Z. (2008). Thylakoid membrane remodeling during state transitions in Arabidopsis. *The Plant Cell*, 20(4), 1029-1039.

Flannery, S. E., Hepworth, C., Wood, W. H., Pastorelli, F., Hunter, C. N., Dickman, M. J., ... & Johnson, M. P. (2021). Developmental acclimation of the thylakoid proteome to light intensity in Arabidopsis. *The Plant Journal*, 105(1), 223-244.

Lichtenthaler, H. K., Burgstahler, R., Buschmann, C., Meier, D., Prenzel, U., & Schönthal, A. (1983). Effect of high light and high light stress on composition, function and structure of the photosynthetic apparatus. In *Effects of Stress on Photosynthesis: Proceedings of a conference held at the 'Limburgs Universitair Centrum' Diepenbeek, Belgium, 22–27 August 1982* (pp. 353-370). Springer Netherlands.

Schumann, T., Paul, S., Melzer, M., Dörmann, P., & Jahns, P. (2017). Plant growth under natural light conditions provides highly flexible short-term acclimation properties toward high light stress. *Frontiers in Plant Science*, 8, 681.

Wood, W. H., Barnett, S. F., Flannery, S., Hunter, C. N., & Johnson, M. P. (2019). Dynamic thylakoid stacking is regulated by LHClI phosphorylation but not its interaction with PSI. *Plant physiology*, 180(4), 2152-2166.

8. Figures 3F and 3G: Labels are missing. What is Ct and what is HL?

We apologize for the omission. We have added the information in Fig. 3F and G, which corresponds to the Ct and HL treatment, respectively.

REVIEWERS' COMMENTS

Reviewer #1 (Remarks to the Author):

The authors have sufficiently revised their manuscript to address this reviewer's questions and comments.

Reviewer #2 (Remarks to the Author):

The authors addressed my concerns and critique adequately.